Full Paper – MIDL 2026

# MultiPersistence Topological Fusion with Vision Transformers for Skin Cancer Detection

**Fulya Tastan**[*1]                                                                FTASTAN@UCSC.EDU
**Sayoni Chakraborty**[*2]                                    SAYONI.CHAKRABORTY@UTDALLAS.EDU
**Sangyeon Lee**[*2]                                               SANGYEON.LEE@UTDALLAS.EDU
**Baris Coskunuzer**[2]                                                  COSKUNUZ@UTDALLAS.EDU

[1] *UC Santa Cruz, Mathematics Department, Santa Cruz, CA 95064 USA*

[2] *UT Dallas, Department of Mathematical Sciences, Richardson, TX 75080 USA*

**Editors:** Accepted for publication at MIDL 2026

## Abstract

Skin cancer is a common and potentially fatal disease where early detection is crucial, especially for melanoma. Current deep learning systems classify skin lesions well, but they primarily rely on appearance cues and may miss deeper structural patterns in lesions. We present TopoCon-MP, a method that extracts multiparameter topological signatures from dermoscopic images to capture multiscale lesion structure, and fuses these signatures with Vision Transformers using a supervised contrastive objective. Across three public datasets, TopoCon-MP improves in-distribution performance over strong pretrained CNN and ViT baselines, and in cross-dataset transfer, it maintains competitive performance. Ablations show that both multiparameter topology and contrastive fusion contribute to these gains. The resulting topological channels also provide an interpretable view of lesion organization that aligns with clinically meaningful structures. Overall, TopoCon-MP demonstrates that multipersistence-based topology can serve as a complementary modality for more robust skin cancer detection.

**Keywords:** Skin lesion classification, dermoscopy, medical image analysis, cubical multiparameter persistence, topological data analysis, supervised contrastive learning

Skin cancer is among the most common and potentially lethal malignancies worldwide, so early and accurate detection is essential for reducing morbidity and mortality. Clinical decision support systems based on machine learning have shown strong promise for automating the analysis of dermoscopic images, with deep models approaching dermatologist-level performance on lesion classification tasks (Codella et al., 2018; Tang et al., 2020). However, conventional CNN and transformer architectures are driven primarily by local pixel and texture cues and may underutilize global properties such as lesion shape, connectivity patterns, and boundary irregularities that often distinguish malignant from benign lesions (Gutman et al., 2016).

Topological data analysis (TDA) offers a complementary perspective by extracting stable, multiscale descriptors of an image's geometric and connectivity structure (Carlsson, 2009). Persistent homology summarizes how features such as connected components and holes appear and disappear across a filtration, yielding signatures that are robust to small deformations and illumination changes. Prior work has demonstrated the value of TDA in medical imaging, including tumor characterization and histopathology (Hofer et al., 2017),

---

[*] Contributed equally

but almost all existing approaches rely on single-parameter filtrations and therefore cannot directly capture how multiple imaging cues interact across scales.

In this paper, we introduce *cubical multiparameter persistence* for dermoscopic image analysis and, to our knowledge, provide the first systematic study of multipersistence for skin cancer detection. We build coupled cubical filtrations that jointly encode intensity, scale, and simple texture surrogates, producing multiparameter Betti summaries that reflect cross-parameter interactions that single-parameter pipelines can miss. To fuse these topological signals with modern vision backbones, we integrate the multipersistence descriptors with a Vision Transformer (ViT) using *supervised contrastive learning*, encouraging class-consistent agreement between image features and topological structure in a shared representation space.

We evaluate our method, TopoCon-MP, on three publicly available skin lesion datasets that include melanoma, basal cell carcinoma, squamous cell carcinoma, and benign nevi. Experiments show that multipersistence alone is competitive with strong pretrained baselines, and that topology-aware contrastive fusion improves AUC, accuracy, and balanced accuracy over both CNN and ViT models, with particularly clear gains in cross-dataset transfer. In addition, the learned topological channels provide interpretable cues that co-localize with clinically meaningful structures such as pigment networks and irregular lesion borders.

Our contributions are:

- We present the first application and systematic evaluation of *cubical multiparameter persistence* for automated skin lesion classification.

- We propose a topology and vision fusion strategy that aligns multipersistence descriptors with a ViT using supervised contrastive learning to obtain joint representations.

- We demonstrate consistent performance gains over strong pretrained CNN and ViT baselines across multiple datasets, including a cross-dataset transfer setting, with ablations showing the advantage of multipersistence over single-parameter cubical persistence.

- We provide qualitative evidence that topological signals highlight clinically relevant morphology, supporting interpretability for dermatological imaging.

## 1. Background

### 1.1. Related Work

**Machine learning methods in skin cancer detection.** Machine learning, in particular deep learning, has transformed automated skin lesion analysis. Esteva et al. showed that convolutional neural networks (CNNs) trained on large collections of clinical and dermoscopic images can reach dermatologist-level performance on lesion classification (Esteva et al., 2017). Follow-up work fine-tuned ImageNet-pretrained CNNs such as ResNet on curated dermoscopy datasets (Menegola et al., 2017), examined robustness across cohorts (Brinker et al., 2019), and introduced benchmarks like HAM10000 that enabled large-scale evaluation and ensemble methods (Tschandl et al., 2018; Codella et al., 2018). More recent studies have explored multimodal models that fuse dermoscopy with patient metadata

for improved risk stratification (Li and Shen, 2020) and transformer-based architectures that capture long-range spatial dependencies (Yuan et al., 2021). Despite this progress, challenges remain around data heterogeneity, interpretability, and deployment in real clinics (Patel et al., 2021). In particular, most models focus on pixel and texture cues and underutilize global properties such as lesion shape, connectivity, and boundary irregularity that are central to dermoscopic diagnosis (Gutman et al., 2016).

**Topological machine learning in medical image analysis.** Topological data analysis (TDA) provides stable, multiscale descriptors of geometric and connectivity structure (Carlsson, 2009). Persistent homology (PH) has been applied in many biomedical settings, including modeling cell development (McGuirl et al., 2020), delineating tumor margins (Qaiser et al., 2019), analyzing brain connectivity (Saggar et al., 2018), and extracting genomic signatures (Lum et al., 2013); see Skaf et al. (Skaf and Laubenbacher, 2022) for a survey. Building on these ideas, topological deep learning integrates PH summaries into trainable models (Hofer et al., 2017; Adams et al., 2017), with reported gains in segmentation (Kahle et al., 2021; Santhirasekaram et al., 2023) and classification (Chachólski et al., 2019; Johnson et al., 2022). Applications to melanoma and skin lesion analysis have begun to appear (Maurya et al., 2024; Chung et al., 2018), but they typically use single-parameter filtrations and do not model interactions between multiple imaging cues.

Multiparameter persistent homology generalizes PH to filtrations indexed by more than one parameter and has been developed theoretically and algorithmically in recent years (Botnan and Lesnick, 2022; Loiseaux et al., 2023; Korkmaz et al., 2025). To our knowledge, it has not yet been explored for dermoscopic image analysis. Our work introduces cubical multiparameter persistence for skin cancer detection and studies both standalone topological models and hybrid models that fuse multipersistence summaries with Vision Transformers. This fills a gap between existing TDA-based approaches, which rely on single-parameter pipelines, and mainstream DL methods, which largely ignore explicit topological structure.

### 1.2. Cubical Persistence

Persistent homology (PH) is a core tool in TDA for extracting multiscale structure from data such as point clouds, networks, and images (Dey and Wang, 2022). We focus on its image variant, *cubical persistence.* A brief overview follows; see (Coskunuzer and Akçora, 2024) for details. PH proceeds in three steps:

- **Filtration**: build a nested sequence of topological spaces.
- **Persistence diagrams**: record feature births and deaths across the sequence.
- **Vectorization**: map diagrams to fixed-length representations for machine learning.

*Step 1. Constructing filtrations.* In images, filtrations are typically cubical. Starting from a grayscale (or single color-channel) image $\mathcal{X} \in \mathbb{R}^{r \times s}$ with pixel values $\gamma_{ij}$ and a sequence of thresholds $t_1 < \cdots < t_N$, we form a sublevel sequence $\mathcal{X}_1 \subset \cdots \subset \mathcal{X}_N$ with $\mathcal{X}_n = \{\Delta_{ij} \subset \mathcal{X} \mid \gamma_{ij} \leq t_n\}$, where

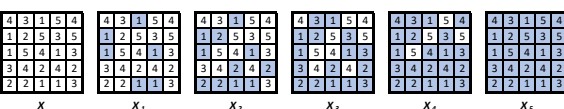

Figure 1: For the $5 \times 5$ image $\mathcal{X}$ with the given pixel values, the sublevel filtration is the sequence of binary images $\mathcal{X}_1 \subset \cdots \subset \mathcal{X}_5$.

$\Delta_{ij}$ is the pixel at position $(i, j)$. Intuitively, pixels become "active" as the threshold increases, yielding a nested family of binary images (Fig. 1).

*Step 2. Persistence diagrams.* PH tracks when topological features appear and disappear along $\{\mathcal{X}_n\}$. If a feature $\sigma$ is born at $t_m$ and disappears at $t_n$ with $m < n$, the pair $(b_\sigma, d_\sigma) = (t_m, t_n)$ is added to the $k$-dimensional diagram $\mathrm{PD}_k(\mathcal{X})$, where $k$ indexes connected components ($k = 0$), holes ($k = 1$), and higher-dimensional cavities. The lifespan $d_\sigma - b_\sigma$ quantifies the prominence of the feature. In Fig. 1, $\mathrm{PD}_0$ captures connected components and $\mathrm{PD}_1$ captures holes.

*Step 3. Vectorization.* Since persistence diagrams are multisets of birth–death pairs, they are not directly suitable for standard learning architectures. Vectorization (Ali et al., 2023) converts diagrams into fixed-length representations. We primarily use *Betti vectors*, where $\beta_k(t_n)$ counts the number of alive $k$-dimensional features at threshold $t_n$, yielding $\overrightarrow{\beta_k} = [\beta_k(t_1), \ldots, \beta_k(t_N)]$. For example, for Fig. 1, $\overrightarrow{\beta_0} = [4, 2, 1, 1, 1]$ and $\overrightarrow{\beta_1} = [0, 1, 2, 2, 0]$. Alternative encodings include persistence images (Adams et al., 2017), landscapes (Bubenik and Dłotko, 2017), silhouettes (Chazal et al., 2014), and kernel methods (Ali et al., 2023). We favor Betti vectors for their efficiency, interpretability, and sequence form, which extend to multiparameter Betti tensors and integrate well with transformer models.

## 2. Methodology

### 2.1. Cubical Multiparameter Persistence

**From single to multiparameter persistence.** In single-parameter cubical persistence, a grayscale (or single-channel) image $\mathcal{X}$ induces a filtration $\{\mathcal{X}_n\}_{n=1}^N$ indexed by thresholds $\{t_n\}$, where $\mathcal{X}_n$ is the binary image obtained by activating pixels with intensity below $t_n$. Persistent homology then tracks when connected components, holes, and higher-dimensional features appear and disappear as the threshold increases, and summarizes them with barcodes or persistence diagrams.

In multiparameter persistence (MP), we let the image evolve along two or more directions at once. We focus on the two-parameter case. A *bifiltration* of an $r \times s$ image $\mathcal{X}$ is a family $\{\mathcal{X}_{m,n}\}$ of binary images such that $\mathcal{X}_{m,n} \subset \mathcal{X}_{m+1,n}$ and $\mathcal{X}_{m,n} \subset \mathcal{X}_{m,n+1}$. Each row and each column is a standard 1D filtration, and together they form a grid of nested images. Applying homology at each grid point gives a collection of topological features that now live over a 2D index set instead of a single line.

A key difference from the single-parameter case is that there is no unique way to assign a single birth and death time to each feature, since the indices $(m, n)$ are only partially ordered. As a result, there is no canonical barcode or persistence diagram in general multiparameter settings (Botnan and Lesnick, 2022). Several alternative summaries have been proposed, such as rank invariants and MP landscapes (Vipond et al., 2021; Loiseaux et al., 2023), but most of them are still relatively heavy for practical large-scale imaging.

**Betti tensors as multiparameter signatures.** In this work we adopt a simple but effective summary based on Betti numbers over the grid. For each grid point $(m, n)$ and each homological dimension $k$, we define

$$\beta_{m,n}^k = \{\text{the count of k-dimensional topological features in } \mathcal{X}_{m,n}\}$$

Collecting these values over the grid yields a 2D *Betti tensor* $\quad \mathbf{B}_k(\mathcal{X}) = [\beta_{m,n}^k] \in \mathbb{N}^{M \times N}$. Intuitively, $\mathbf{B}_0(\mathcal{X})$ records how the number of connected components changes across two parameters, and $\mathbf{B}_1(\mathcal{X})$ does the same for holes. Unlike persistence diagrams, Betti tensors

do not distinguish long-lived from short-lived features, but they provide a compact, grid-aligned representation that is easy to store and to feed into neural networks. This type of Betti-based encoding has been empirically effective in several medical and histopathological imaging tasks (Qaiser et al., 2019; Yadav et al., 2023; Du et al., 2022). Additional details and a more formal connection to the multipersistence literature are given in Appendix A.

**Color multifiltrations for dermoscopic images.** RGB dermoscopic images naturally support multiparameter filtrations (Korkmaz et al., 2025). Let $\mathcal{X}$ be an RGB image with channel values $R_{ij}, G_{ij}, B_{ij} \in [0, 255]$ for each pixel (cubical cell) $\Delta_{ij}$. In general, choosing threshold sets $\{s_m\}_{m=1}^{N_1}$, $\{t_n\}_{n=1}^{N_2}$, $\{v_r\}_{r=1}^{N_3}$ for the three channels defines a three-parameter multifiltration $\mathcal{X}_{m,n,r} = \{\Delta_{ij} \subset \mathcal{X} \mid R_{ij} \leq s_m, G_{ij} \leq t_n, B_{ij} \leq v_r\}$, and corresponding 3D Betti tensors $[\beta_{m,n,r}^k] \in \mathbb{N}^{N_1 \times N_2 \times N_3}$. Figure 2 shows a small toy example with a $3 \times 3$ grid for clarity.

For our experiments we adopt a computationally efficient two-parameter specialization tailored to dermoscopic images. We construct a bifiltration over the red and green channels (identified as most informative on a validation set) using $M = N = 20$ thresholds to form a $20 \times 20$ grid. For each image we compute the corresponding $\beta_0$ and $\beta_1$ tensors together with activated-pixel counts, and stack them into a $3 \times 20 \times 20$ *topological image*. This multipersistence representation is used both as input to XGBoost baselines and as the topological branch in the TopoCon-MP fusion model. A more formal discussion of multiparameter persistence, barcode obstructions, and alternative summaries (including our Betti tensor view) is provided in Appendix A.

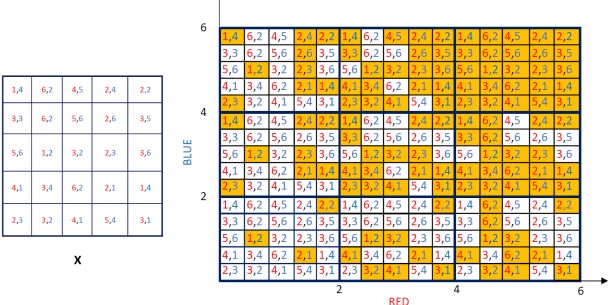

Figure 2: **Toy example.** For an image $\mathcal{X}$ with two color channels, a simple color bifiltration produces a $3 \times 3$ grid of binary images (the actual grid we used is $20 \times 20$). Horizontally, pixels are activated (colored orange) when their red value falls below the threshold, and vertically, activation depends on the blue value. Each row and column forms an ordinary one dimensional filtration, while the grid as a whole defines a two dimensional multiparameter filtration.

### 2.2. Topology Aware Supervised Contrastive Learning

Supervised contrastive learning encourages representations that cluster samples from the same class while separating those from different classes. Standard supervised contrastive methods typically construct multiple "views" of each image using random augmentations such as cropping, rotation, or intensity jitter. In dermoscopy, however, aggressive spatial augmentations can distort lesion boundaries or alter diagnostically relevant texture patterns.

We therefore propose a *topology aware* supervised contrastive framework that uses the original dermoscopic image and its multiparameter topological embedding as two semantically consistent views of the same case (see Fig. 3). For each input image $I$, we compute its cubical multiparameter persistence representation, capturing structural and morphological characteristics in a label preserving and anatomically coherent way. The resulting bifiltration produces a 2D topological image $\Psi(I) \in \mathbb{R}^{H \times W \times 3}$, where the three channels correspond

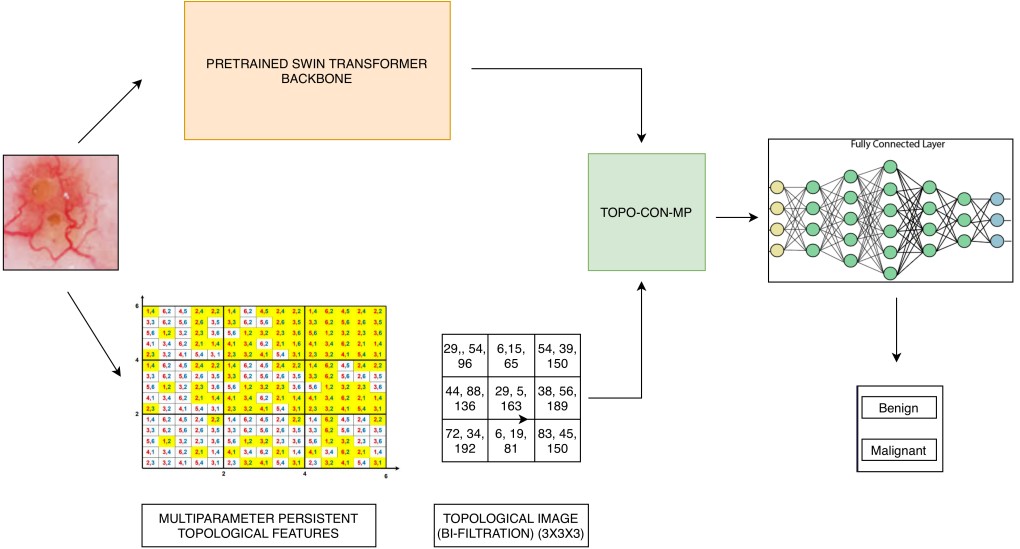

Figure 3: **Overview of TopoCon-MP.** The raw dermoscopic image is processed by a pretrained Swin Transformer backbone to obtain semantic image features. In parallel, we compute multiparameter Betti tensors on a fixed grid and stack $\beta_0$, $\beta_1$, and activated-pixel counts into a $3 \times 20 \times 20$ topological image. This topological image is encoded with an MLP and aligned with the Swin features via a topology-aware supervised contrastive loss. The fused representation is passed to a final classifier for benign vs. malignant prediction.

to $\beta_0$, $\beta_1$, and the activated pixel map derived from the multipersistence computation. This RGB style topological image emphasizes global topology and boundary structure without introducing augmentation induced bias, which is a common issue in medical imaging where strong contrastive augmentations (e.g., heavy color jitter, aggressive cropping, blur) can distort clinically meaningful cues and produce label-inconsistent views, encouraging invariances that are undesirable for diagnosis.

An image encoder $f_\theta(\cdot)$ (a pretrained Swin Transformer backbone with a linear head) and a topology encoder $g_\phi(\cdot)$ (an MLP on $\Psi(I)$) produce latent embeddings $z_I = f_\theta(I), z_T = g_\phi(\Psi(I))$. These embeddings are concatenated and fed to a classifier for the main lesion classification task. In parallel, they are mapped through projection heads and used in a supervised contrastive loss: samples that share the same class label, whether they come from the image branch or the topology branch, are treated as positives, and samples from different classes are treated as negatives (following the formulation of Khosla et al.).

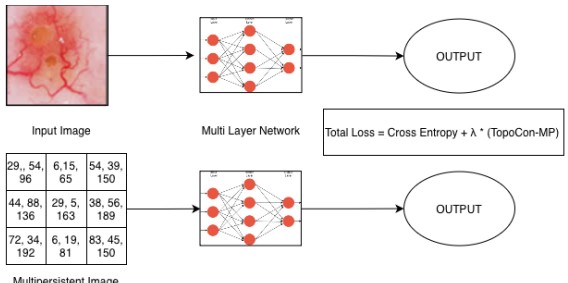

Figure 4: **Supervised contrastive framework.** A dermoscopic image and its multipersistence image (toy $3 \times 3 \times 3$ example) are encoded by separate networks. Their embeddings are used for classification and a supervised contrastive loss that aligns image and topology representations.

The final training objective combines cross entropy and supervised contrastive losses, $\mathcal{L}_{\text{total}} = \mathcal{L}_{\text{CE}} + \lambda \mathcal{L}_{\text{SupCon}}$, where $\lambda$ bal-

ances discriminative and alignment terms (See Fig. 4). This objective aligns image and topology embeddings in a class consistent manner while preserving classification performance. By using topological representations as label preserving views instead of only random augmentations, TopoCon-MP provides contrastive supervision better tailored to limited data and anatomy sensitive medical imaging.

## 3. Experiments

**Datasets.** We evaluate our framework on three publicly available dermoscopic image datasets: DermaMNIST, MILK-10K, and PAD-UFES-20. Together they cover a range of lesion types, acquisition devices, and class distributions. All images are expert annotated der-

Table 1: Summary of datasets.

| Dataset | # Images | # Classes |
|---------|----------|-----------|
| DermaMNIST | 10,015 | 7 |
| MILK-10K | 5,240 | 11 |
| PAD-UFES-20 | 2,298 | 6 |

moscopic RGB scans, resized to a fixed resolution before training. DermaMNIST (Yang et al., 2023), derived from the HAM10000 dataset (Tschandl et al., 2018) from the ISIC 2018 challenge (Codella et al., 2019), contains 10,015 dermoscopic images across seven diagnostic categories and is a standard skin lesion classification benchmark. MILK-10K (Philipp et al., 2025) is a collection of approximately 10,000 dermoscopic and clinical images across eleven classes. To keep the setting consistent across datasets, we use only the dermoscopic subset (5,240 images) in our experiments. PAD-UFES-20 (Pacheco et al., 2020) comprises 2,298 smartphone-acquired dermoscopic images labeled by pathologists into six diagnostic categories, introducing realistic variability in acquisition conditions and illumination.

**Data splits and cross dataset evaluation.** For each dataset we perform an 80:10:10 split into training, validation, and test sets (grouped by patient identifier where available to avoid leakage). We first train and evaluate models on each dataset independently. Furthermore, to assess **cross-dataset generalization**, we also conduct a transfer experiment: models are trained on the five lesion classes common to DermaMNIST and the dermoscopic subset of MILK-10K, and evaluated on the corresponding subset of PAD-UFES-20.

**Preprocessing and Topological Features.** All dermoscopic images are resized to $224 \times 224$ pixels to ensure uniform spatial resolution across datasets. Standard normalization is applied to the RGB channels before feature extraction.

*Single Persistence Topological Descriptors.* For the single persistent homology setting, we compute Betti features using 50 filtration thresholds ($n\_bins = 50$). For each image, this produces 50-dimensional vectors for Betti-0, Betti-1, and the number of activated pixels, computed independently across the RGB and grayscale channels. This results in a total of 4 (channels) $\times$ 3 (feature types) $\times$ 50 = 600 topological features per image.

*MultiPersistence Topological Descriptors.* To capture richer structural interactions, we further compute cubical multi-parameter persistence using bifiltration over the red and green channels with 20 thresholds each. This yields a $20 \times 20$ bifiltration grid for each image, producing a 2D topological map with three channels corresponding to Betti-0, Betti-1, and activated pixels calculated during bifiltration. When flattened, this representation provides $20 \times 20 \times 3 = 1200$ multi-parameter topological features per image.

These topological features are first evaluated independently using XGBoost classifiers to assess their discriminative capability. Subsequently, the multi-parameter topological maps

are integrated into the **TopoCon-MP** framework, where they are combined with Swin Transformer embeddings for supervised contrastive learning and classification.

**Hyperparameters.** We used the Adam optimizer with a learning rate of $1e^{-4}$, batch size of 64, and cross-entropy loss with class weights for all experiments. All images were resized to $224\times224$ and normalized with the ImageNet mean and standard deviation. Baseline CNN and ViT models, and our TopoCon-MP model were trained for 15 epochs without augmentation, using ImageNet-pretrained backbones with frozen weights while keeping BatchNorm statistics and Dropout layers active.

For the transfer learning setup (training on MILK-10K and DermaMNIST, testing on PAD-UFES-20), we used identical hyperparameters and applied early stopping based on validation macro-F1 (patience = 3).

Our contrastive fusion model (TopoCon-MP) was trained with the AdamW optimizer (weight decay of $1e^{-2}$) for 30 epochs under cosine-annealing scheduling. The model fused Swin-T image embeddings with $3 \times 20 \times 20$ topological maps derived from 1200-dimensional multi-persistence vectors. Each vector was normalized in 400-dimensional blocks and reshaped to a $3 \times 20 \times 20$ tensor. The fusion module consisted of LayerNorm, a linear layer, ReLU activation, and a dropout rate of 0.3. The projection head for contrastive learning was a two-layer MLP with dimensions $768 \rightarrow 256 \rightarrow 64$, LayerNorm and ReLU activations, mapping fused embeddings to a 64-dimensional latent space. We used a supervised contrastive loss with $\lambda = 0.1$ and temperature $\tau = 0.07$, combined with cross-entropy loss for classification. All models used AMP, TF32, and gradient clipping at 1.0. The best model was selected based on the highest validation AUC. Multipersistence feature extraction on the DermaMNIST dataset was performed on an HPC system using single-core CPU execution. For $224\times224$ images, computing the $3\times20\times20$ (Betti-0, Betti-1, activated pixels) tensor requires 0.229 s per image on average, corresponding to an overall preprocessing time of approximately 38 minutes for the full DermaMNIST dataset (10,015 images).Training TopoCon-MP with a frozen Swin-T backbone on the DermaMNIST dataset required approximately 55 minutes on an HPC system using a single NVIDIA A100 GPU. Our code is available at https://github.com/sayoni-c98/MIDL2026-TopoConMP

**Baselines.** We compare our approach against widely used 2D convolutional and transformer-based architectures. For convolutional baselines, we include MobileNetV3-Large-100 (Howard et al., 2019), DenseNet121 (Huang et al., 2017), ResNet50 (He et al., 2016), and EfficientNetV2-S (Tan and Le, 2021), representing compact, densely connected, residual, and compound-scaled network families respectively. For transformer-style baselines, we evaluate ViT-B/16 (Dosovitskiy et al., 2021), MobileViT-S (Mehta and Rastegari, 2022), and Swin-T (Liu et al., 2021), covering both pure and hybrid vision transformer designs. All models use ImageNet pretrained weights and are fine tuned on the dermoscopic datasets with identical optimization and augmentation settings.

**Results.** Table 2 reports the performance of all CNN, transformer, and TopoCon-MP models on the three dermoscopic datasets. On every dataset, TopoCon-MP attains the best AUC, accuracy, and macro F1, and generally improves sensitivity and specificity as well. This suggests that fusing multiparameter topological features with a supervised contrastive objective yields more discriminative representations than image-only baselines. On DermaMNIST and MILK-10K, absolute sensitivity remains modest for all methods due

Table 2: **Baseline comparison.** We compare TopoCon-MP with strong pretrained CNN and ViT models. The **best**, second, and third scores in each column are highlighted.

| Model | DermaMNIST | | | | | MILK-10K | | | | | PAD-UFES-20 | | | | |
|---|---|---|---|---|---|---|---|---|---|---|---|---|---|---|---|
| | AUC | Acc. | F1 | Sens. | Spec. | AUC | Acc. | F1 | Sens. | Spec. | AUC | Acc. | F1 | Sens. | Spec. |
| MobileNetV3 | 88.5 | 63.4 | 43.3 | 50.0 | 93.1 | 81.9 | 50.6 | 29.8 | 32.3 | 94.6 | 82.7 | 52.0 | 42.7 | 45.5 | 89.9 |
| DenseNet121 | 87.4 | 59.4 | 38.5 | 50.2 | 93.0 | 77.3 | 45.4 | 25.2 | 28.6 | 94.1 | 78.2 | 47.4 | 40.8 | 44.8 | 88.6 |
| ResNet50 | 84.2 | 61.1 | 38.2 | 45.9 | 92.8 | 79.9 | 47.9 | 20.7 | 22.4 | 94.0 | 74.2 | 50.3 | 35.0 | 35.7 | 89.2 |
| EfficientNetV2-S | 82.5 | 56.3 | 36.0 | 43.9 | 91.9 | 73.7 | 42.9 | 27.1 | 34.0 | 93.9 | 75.1 | 43.4 | 37.4 | 41.3 | 88.2 |
| ViT-B/16 | 93.7 | 75.7 | 61.5 | 61.6 | 94.5 | 79.4 | 55.3 | 35.9 | 37.8 | 94.9 | 87.7 | 66.5 | 56.4 | 55.6 | 93.2 |
| MobileViT-S | 89.1 | 68.9 | 54.0 | 61.8 | 94.3 | 82.5 | 52.1 | 34.3 | 38.1 | 94.9 | 87.7 | 64.7 | 48.0 | 49.6 | 92.1 |
| Swin-T | 93.3 | 73.7 | 57.1 | 63.0 | 94.4 | 87.7 | 69.3 | 47.5 | 46.9 | 96.2 | 88.6 | 75.7 | 60.3 | 59.8 | 94.6 |
| TopoCon-MP | 94.9 | 79.3 | 62.2 | 59.7 | 94.3 | 94.5 | 71.0 | 48.9 | 50.4 | 96.7 | 93.0 | 77.8 | 73.0 | 72.0 | 95.0 |

to class imbalance and ambiguous cases at a fixed decision threshold, but TopoCon-MP achieves the highest sensitivity while also improving AUC and macro-F1, indicating gains are not driven by a specificity-only tradeoff.

To assess cross-dataset generalization, we train all models on the five shared classes from DermaMNIST and the dermoscopic subset of MILK-10K and evaluate on the corresponding subset of PAD-UFES-20 (Table 3). In this transfer setting, ViT-B/16 achieves the highest AUC and F1, while TopoCon-MP remains competitive: its AUC is within 0.4 of ViT-B/16, it obtains the second-best accuracy, and it consistently outperforms the CNN baselines across metrics. These results indicate that multipersistence features help stabilize performance under domain shift, even though our current fusion does not yet surpass the strongest transformer.

**Ablation studies.** We conduct two ablations to disentangle the contributions of the topological representation and the fusion strategy. First, in Table 4 we compare single parameter (SP) cubical persistence on grayscale and on all RGB channels with our multiparameter red plus green (MP RG) encoding, using the same XGBoost classifier. On DermaMNIST, SP across all channels attains the best AUC and accuracy, indicating that a simple multi channel filtration already captures most of the useful topology on this relatively clean benchmark. On MILK-10K, MP RG yields the highest AUC, and on the more heterogeneous PAD UFES 20 dataset it substantially improves accuracy, F1, sensitiv-

Table 3: **Cross dataset transfer to PAD-UFES-20.** Models are trained on the five shared classes from DermaMNIST and the dermoscopic subset of MILK-10K and evaluated on PAD-UFES-20 without fine tuning. For each metric, the **best**, second, and third scores across models are highlighted.

| Model | AUC | Acc | F1 | Sens | Spec |
|---|---|---|---|---|---|
| MobileNetV3 | 66.2 | 39.3 | 30.9 | 32.5 | 82.8 |
| DenseNet121 | 66.6 | 36.6 | 30.3 | 31.6 | 81.7 |
| ResNet50 | 64.0 | 39.8 | 20.8 | 24.7 | 80.6 |
| EfficientNetV2-S | 62.4 | 34.9 | 27.2 | 29.3 | 82.4 |
| ViT-B/16 | 78.9 | 51.3 | 45.5 | 46.8 | 86.2 |
| MobileViT-S | 73.8 | 41.1 | 33.9 | 38.8 | 83.8 |
| Swin-T | 78.7 | 47.6 | 42.4 | 46.3 | 86.1 |
| TopoCon-MP | 78.5 | 50.3 | 36.7 | 42.9 | 85.5 |

ity, and specificity over SP variants, suggesting that MP becomes more beneficial as acquisition conditions and lesion appearance vary.

Second, Table 5 evaluates different ML classifier models on the same $3 \times 20 \times 20$ multipersistence tensors. XGBoost provides a strong topology only baseline, while a applying

2D CNN on these MP outputs (MP+CNN) performs worse and is unstable on MILK 10K and PAD UFES 20. In contrast, our full TopoCon-MP model (MP+SupCon), which jointly trains image and topology encoders with supervised contrastive alignment, delivers large and consistent gains in AUC, accuracy, F1, sensitivity, and specificity across all datasets. These results indicate that both the multiparameter features and the topology aware contrastive fusion are important for the final performance.

Table 4: **Topological Feature Ablation.** The results of our ablation study of XGBoost models on topological features across different channels and datasets.

| | | DermaMNIST | | | | | MILK-10K | | | | | PAD-UFES-20 | | | | |
|---|---|---|---|---|---|---|---|---|---|---|---|---|---|---|---|---|
| TDA model | # Feat. | AUC | Acc. | F1 | Sens. | Spec. | AUC | Acc. | F1 | Sens. | Spec. | AUC | Acc. | F1 | Sens. | Spec. |
| SP-Grayscale | 150 | 84.8 | 70.2 | 32.9 | 29.0 | 89.8 | 72.6 | 58.9 | 19.0 | 19.2 | 93.7 | 74.0 | 46.5 | 27.8 | 28.4 | 86.7 |
| SP-All channels | 600 | 92.8 | 75.3 | 49.0 | 42.8 | 92.1 | 74.9 | 59.5 | 19.6 | 19.8 | 93.8 | 76.2 | 47.8 | 28.6 | 29.5 | 86.9 |
| MP-Red and Green | 1200 | 87.1 | 72.0 | 31.0 | 29.2 | 90.7 | 79.8 | 56.0 | 19.0 | 19.1 | 93.4 | 74.2 | 57.0 | 38.0 | 39.1 | 89.3 |

**Limitations.** While we focus on dermoscopic images, where color and lesion structure are relatively standardized, extending the approach to other modalities with different artifacts and intensity statistics may require redesigning the filtration and preprocessing for stability, and the current multipersistence grid may not be optimal. Future work will study filtration sensitivity and develop modality specific, acquisition robust filtrations.

**Visualization.** To better understand what the multipersistence descriptors capture, we visualize them on MILK-10K and DermaMNIST. In particular, Appendix C provides two complementary views for MILK-10K: Figure 7 shows mean $\beta_0$ and $\beta_1$ Betti curves with confidence bands across color channels, while Figure 8 displays classwise median red and green Betti tensors and activated pixel maps. Together, these plots reveal consistent, class specific patterns in multiscale topology, for example broader peaks and shifted hotspots for melanoma and keratinocytic lesions, providing qualitative evidence that our multipersistence features capture clinically meaningful lesion structure rather than arbitrary handcrafted cues. Figure 5 shows analogous DermaMNIST red-channel $\beta_0$ curves, revealing class-specific differences in connected-component evolution, including shifts in peak intensity thresholds as well as changes in peak magnitude and decay rate as the sublevel set grows.

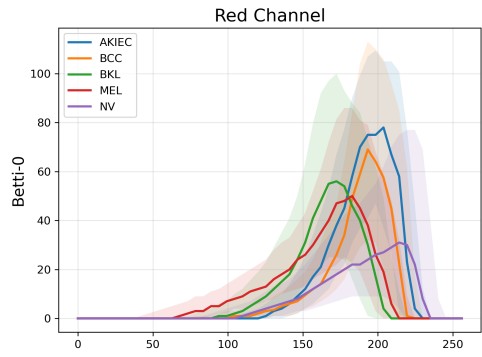

Figure 5: **DermaMNIST Betti-0 curves.** Mean $\beta_0$ curves with 40% confidence bands for DermaMNIST lesion classes (AKIEC, BCC, BKL, MEL, NV) as a function of red-channel intensity threshold, showing class-specific patterns in the evolution of connected components.

## 4. Conclusion

In summary, we introduced cubical multiparameter persistence for dermoscopic image analysis and showed that topology provides complementary signal to modern vision backbones for

Table 5: **ML Ablation.** The performances of different ML models utilizing our $3\times20\times20$ multipersistence outputs. MP-SupCon row represents our TopoCon-MP model.

| TDA features | DermaMNIST | | | | | MILK-10K | | | | | PAD-UFES-20 | | | | |
|---|---|---|---|---|---|---|---|---|---|---|---|---|---|---|---|
| | AUC | Acc. | F1 | Sens. | Spec. | AUC | Acc. | F1 | Sens. | Spec. | AUC | Acc. | F1 | Sens. | Spec. |
| MP+XGBoost | 87.1 | 72.0 | 31.0 | 29.2 | 90.7 | 79.8 | 56.0 | 19.0 | 19.1 | 93.4 | 74.2 | 57.0 | 38.0 | 39.1 | 89.3 |
| MP+CNN | 79.3 | 68.5 | 18.9 | 18.5 | 87.7 | 77.9 | 55.5 | 20.4 | 20.8 | 93.7 | 71.9 | 50.6 | 31.8 | 33.6 | 87.5 |
| MP+SupCon | **94.9** | **79.3** | **62.2** | **59.7** | **94.3** | **94.5** | **71.0** | **48.9** | **50.4** | **96.7** | **93.0** | **77.8** | **73.0** | **72.0** | **95.0** |

skin cancer classification. Across multiple public datasets, multipersistence alone was competitive with strong pretrained baselines, and when aligned with a Vision Transformer via supervised contrastive learning it improved in-distribution performance over CNN and ViT models and, under cross-dataset transfer, achieved competitive AUC. Ablations confirmed that multiparameter topology yields richer cues than single-parameter cubical persistence and that contrastive alignment is important for effective fusion. Beyond performance, we provide concrete qualitative examples where the predicted class can be inspected together with the corresponding Betti curves and multipersistence maps, offering an interpretable auxiliary view of lesion structure that supports model auditing and error analysis. Limitations include the need to choose filtration parameters and grids, the computational overhead of multipersistence, and evaluation focused primarily on dermoscopy. Future work will explore adaptive and differentiable multiparameter filtrations, tighter end-to-end training with vision transformers.

## Acknowledgments

This work was partially supported by National Science Foundation under grants DMS-2220613, and DMS-2229417. The authors acknowledge the Texas Advanced Computing Center (TACC) at UT Austin for providing computational resources that have contributed to the research results reported within this paper.

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

# Appendix

## Appendix A. Multiparameter Persistence

Multiparameter persistence has attracted growing interest for its potential to enrich standard persistent homology. In principle, a multidimensional filtration with several parameters should yield richer topological summaries for machine learning than a one-parameter filtration. However, key technical obstacles have limited its practical impact.

In single-parameter persistence, the threshold space $\{\alpha_i\}$ is totally ordered, so each topological feature in the filtration $\{\Delta_i\}$ has well defined birth and death times. This makes it possible to decompose the associated persistence module $M = \{H_k(\Delta_i)\}_{i=1}^N$ into a multiset of intervals (barcodes) via a structure theorem (Botnan and Lesnick, 2022), which underlies persistence diagrams. For two or more parameters, the threshold set $\{(\alpha_i, \beta_j)\}$ is only partially ordered. Birth and death times are no longer uniquely defined, the one dimensional decomposition theorem does not extend (Botnan and Lesnick, 2022), and barcode representations typically fail to exist or are difficult to describe in a finite way. As a result, a direct barcode style generalization of single-parameter persistence is usually not available, and the classification and invariants of multiparameter modules remain an active area of research in commutative algebra (Eisenbud, 2013).

Despite these challenges, several slicing based methods have been proposed to make use of multiparameter filtrations (Lesnick, 2015; Carrière and Blumberg, 2020; Botnan and Lesnick, 2022). These approaches analyze one dimensional slices of the multiparameter grid, compute standard persistence diagrams along each slice, and then aggregate the resulting diagrams into vectorized summaries. While effective in some settings, they face two main limitations: the summaries can depend strongly on the choice of slicing directions, and compressing information from many diagrams into a low dimensional representation may introduce substantial information loss; see (Botnan and Lesnick, 2022) for a detailed overview.

In this work we adopt a different strategy that avoids slicing. We work directly with the Betti numbers on the grid: for each grid point and each homological dimension $k$, we record the rank of $H_k$ and collect these ranks into Betti tensors. This can be viewed as evaluating the Hilbert function of the underlying multiparameter module on a fixed finite grid. The resulting Betti tensors provide a simple, grid aligned summary that is easy to compute and to feed into neural networks, and they have been empirically effective in several imaging applications (Qaiser et al., 2019; Yadav et al., 2023; Du et al., 2022; Ali et al., 2023).

**Other bifiltration examples.** A well known limitation of grayscale sublevel filtrations is their inability to encode the *size* of topological features; they only reflect differences in function values between the birth and death of a feature. For example, consider a grayscale image where all pixels have intensity 0 except for a single central pixel with intensity 255. The resulting persistence diagram contains a single long bar $[0, 255)$, even though the corresponding hole has diameter 1. Conversely, a binary image $\mathcal{X}_{100}$ might contain a large hole of diameter 20 whose pixels have intensities in $[101, 105]$, so the hole is completely filled by $\mathcal{X}_{105}$. Despite the dramatic change in geometric size, the grayscale sublevel filtration produces only a short bar $(100, 105)$, encoding the contrast but not the spatial scale of the hole.

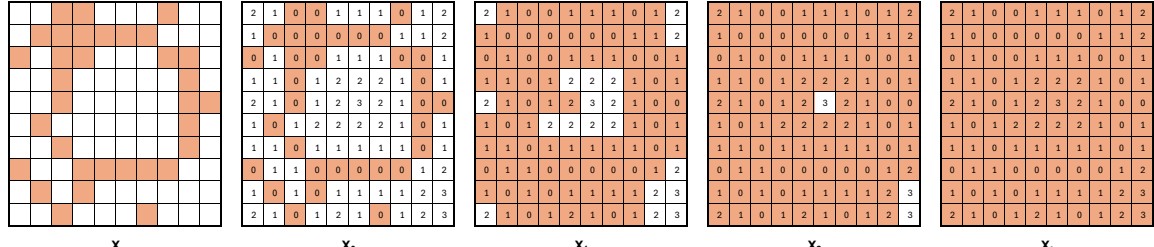

Figure 6: **Erosion filtration.** For a given binary image $\mathcal{X}$, we first define the erosion function (shown in $\mathcal{X}_0$). We then obtain a filtration of binary images $\mathcal{X}_0 \subset \mathcal{X}_1 \subset \cdots \subset \mathcal{X}_3$ by activating pixels that reach the threshold value.

In other words, while persistent homology identifies which topological features appear in a filtration, standard sublevel filtrations do not, by themselves, capture their geometric size. To address this, alternative filtrations such as *erosion, dilation*, and *signed-distance based filtrations* have been proposed (Garin and Tauzin, 2019). These constructions explicitly incorporate scale information and thus complement grayscale sublevel filtrations. In particular, one can combine a grayscale or color channel with an erosion or distance based filtration to obtain meaningful multiparameter persistence signatures for images.

**Why Betti tensors for multipersistence.** Multiparameter persistence offers a rich structural summary, but in practice *vectorization is a primary bottleneck*: many MP representations that retain lifetime information (e.g., signed-barcode measures) can be expensive, sensitive to design choices, and difficult to integrate with standard deep pipelines at scale. We therefore adopt Betti tensors as a deliberate tradeoff between expressivity and learnability. Betti tensors preserve the intrinsic 2D bifiltration geometry as an *image-like* object, enabling simple encoders and contrastive fusion to exploit local and global patterns in the $(r, g)$ filtration grid, whereas more expressive MP vectorizations often collapse this grid structure into an unordered feature set. This choice is also motivated by the data regime: methods with learnable filtrations or heavy MP representation learning (e.g., CuMPerLay-style models (Carrière et al., 2020; Korkmaz et al., 2025)) typically benefit from substantially larger datasets and careful tuning, while our goal is a lightweight, stable MP descriptor that works reliably with limited medical data. Finally, as illustrated by the Betti curve visualizations (Fig. 7), the discriminative signal appears distributed across many small topological events (curve density over thresholds) rather than a few dominant features, which is naturally captured by Betti curves and their tensorized MP extension.

## Appendix B. Sensitivity Analysis

We provide two tables provide a small sensitivity analysis of our topological descriptors with respect to filtration discretization and channel selection. Table 6 varies the number of intensity thresholds used to discretize single parameter persistence on MILK 10K while keeping the downstream classifier fixed. Increasing the resolution from 50 to 100 to 250 thresholds changes the feature dimensionality from 600 to 1200 to 3000, yet the perfor-

Table 6: Sensitivity to threshold resolution for single-parameter persistence features on MILK-10K (all channels).

| Dataset | #thresholds | #features (all channels) | AUC | Acc | F1 | Sens | Spec |
|---|---|---|---|---|---|---|---|
| MILK-10K | 50 | $50 \times 3 \times 4 = 600$ | 74.9 | 59.5 | 19.6 | 19.8 | 93.8 |
| MILK-10K | 100 | $100 \times 3 \times 4 = 1200$ | 74.8 | 60.1 | 20.4 | 20.4 | 94.0 |
| MILK-10K | 250 | $250 \times 3 \times 4 = 3000$ | 74.5 | 60.4 | 21.5 | 21.2 | 94.0 |

Table 7: Single-parameter persistence (SP) features using a single color channel (50 thresholds; 150 features).

| TDA model | #Features | DermaMNIST | | | | | MILK-10K | | | | | PAD | | | | |
|---|---|---|---|---|---|---|---|---|---|---|---|---|---|---|---|---|
| | | AUC | Acc | F1 | Sens | Spec | AUC | Acc | F1 | Sens | Spec | AUC | Acc | F1 | Sens | Spec |
| SP_Red | 150 | 83.4 | 70.0 | 28.2 | 25.6 | 88.9 | 71.5 | 57.4 | 17.3 | 18.0 | 93.4 | 71.1 | 44.7 | 23.5 | 25.0 | 86.2 |
| SP_Green | 150 | 85.4 | 71.2 | 31.6 | 28.5 | 89.9 | 75.6 | 57.6 | 17.9 | 18.3 | 93.4 | 72.3 | 47.8 | 27.7 | 29.0 | 87.0 |
| SP_Blue | 150 | 87.7 | 71.0 | 35.6 | 31.6 | 90.5 | 72.3 | 57.6 | 17.6 | 18.3 | 93.6 | 73.7 | 51.3 | 31.4 | 32.2 | 87.8 |

mance remains stable, with only modest fluctuations in AUC, accuracy, F1, sensitivity, and specificity. This indicates that the extracted TDA signal is not overly dependent on a particular threshold granularity once a reasonable resolution is used. Table 7 reports single channel single parameter persistence features computed from the red, green, or blue channel (same feature budget) across DermaMNIST, MILK 10K, and PAD. The relative ordering across channels differs by dataset, but overall performance is comparable, suggesting that no single channel is universally dominant and motivating our use of complementary channels in the multipersistence construction. Together, these results support that the proposed topological features are reasonably robust to practical choices in filtration discretization and channel selection.

## Appendix C.  Visualizations and Interpretability of Topological Descriptors

To better understand what information our multipersistence descriptors capture, we visualize classwise Betti curves and median multipersistence heatmaps for the MILK-10K dataset.

**Betti curves across color channels.** Tables 7, 8, 9 show the mean Betti curves with 40% confidence bands for five lesion classes (BCC, NV, BKL, MEL, AKIEC) across red, green, blue, and grayscale filtrations. The top row plots $\beta_0$ (number of connected components) as a function of the intensity threshold and the bottom row plots $\beta_1$ (number of holes). The solid (top) and dashed (bottom) lines denote the classwise means, while the shaded regions mark the central 40% of subjects for each class.

Several consistent patterns emerge. First, BKL and AKIEC lesions tend to exhibit higher and broader $\beta_0$ and $\beta_1$ peaks, reflecting a larger number of small islands and holes across intermediate thresholds. This is compatible with their irregular, mottled pigmentation patterns in dermoscopy. In contrast, NV lesions show lower-amplitude curves and

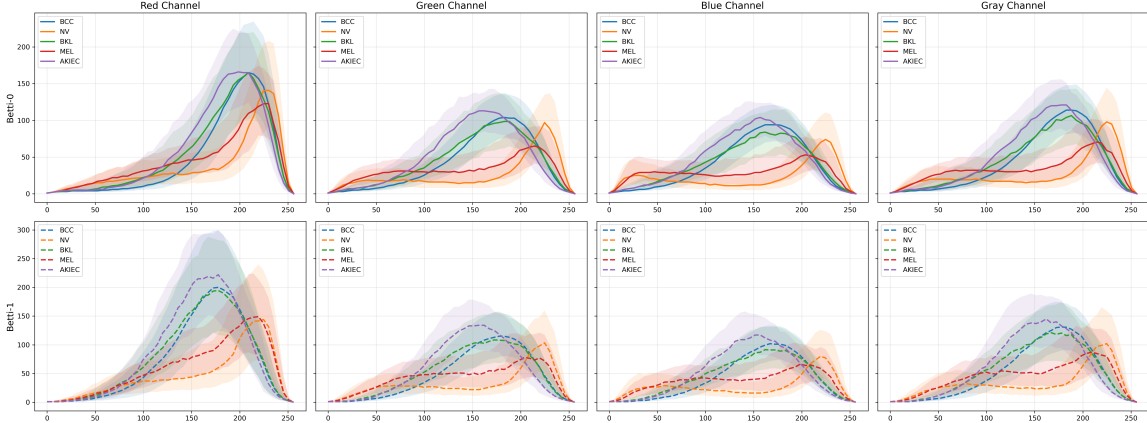

Figure 7: **Betti curves for MILK-10K.** Mean Betti curves with 40% confidence bands for five MILK-10K lesion classes (BCC, NV, BKL, MEL, AKIEC) across color channels. Columns correspond to red, green, blue, and grayscale intensity filtrations. The top row shows $\beta_0$ (number of connected components) and the bottom row shows $\beta_1$ (number of holes) as functions of the threshold value. Solid (top) and dashed (bottom) lines indicate classwise means, and shaded regions mark the central 40% of subjects per class, revealing systematic differences in multiscale topology between lesion types.

narrower peaks, consistent with more homogeneous, compact nevi. MEL curves typically peak at slightly darker thresholds than NV, suggesting richer structure in darker pigment regions, which is in line with the irregular networks and focal globules often seen in melanoma. Across channels, the red and grayscale filtrations yield the strongest separation between classes, which motivated our choice of red–green bifiltrations for the main multipersistence pipeline.

**Multipersistence heatmaps.** While Betti curves summarize topology along a single intensity axis, our model uses bifiltrations over red and green channels. Figure 11 displays classwise median multipersistence heatmaps on the $20 \times 20$ red–green grid. Each row corresponds to one lesion class, and the three columns show $\beta_0$, $\beta_1$, and activated-pixel counts, respectively. Color encodes the median value at each grid point, so bright regions indicate parameter ranges where many connected components, holes, or pixels are present.

These heatmaps reveal complementary structure that is not visible from single-channel curves alone. For example, NV lesions concentrate most of their $\beta_0$ and $\beta_1$ mass in a relatively compact block of lighter thresholds, indicating uniform pigmentation with limited cross-channel variation. BKL and AKIEC classes show broader, more diffuse high-intensity regions that extend toward darker red and green levels, consistent with heterogeneous pigmentation and scattered foci. MEL lesions exhibit a shift of the $\beta_1$ hotspot toward darker-red / mid-green thresholds, suggesting complex hole patterns in specific color combinations that align with irregular pigment networks and streaks. The activated-pixel maps further highlight differences in overall lesion occupancy in the red–green plane, which our model uses jointly with $\beta_0$ and $\beta_1$.

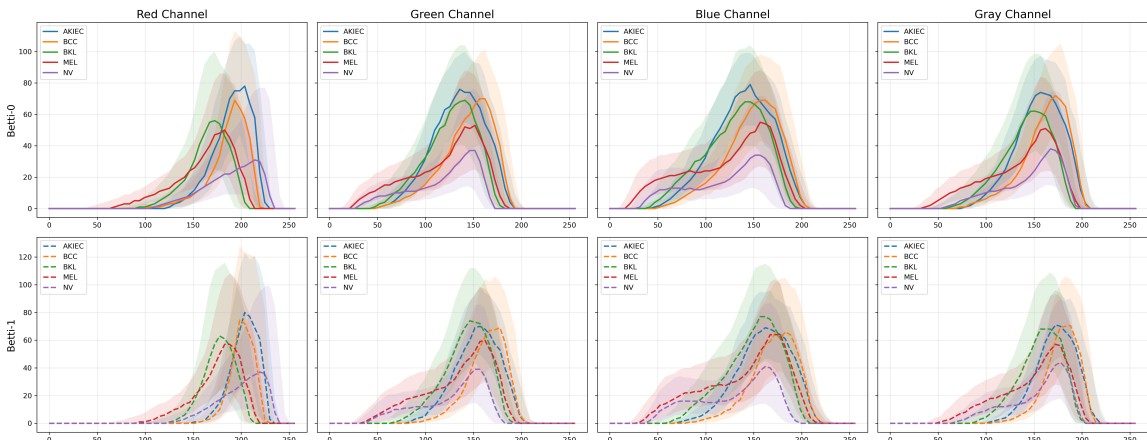

Figure 8: **Betti curves for DermaMNIST.** Mean Betti curves with 40% confidence bands for five MILK-10K lesion classes (BCC, NV, BKL, MEL, AKIEC) across color channels. Columns correspond to red, green, blue, and grayscale intensity filtrations. The top row shows $\beta_0$ (number of connected components) and the bottom row shows $\beta_1$ (number of holes) as functions of the threshold value. Solid (top) and dashed (bottom) lines indicate classwise means, and shaded regions mark the central 40% of subjects per class, revealing systematic differences in multiscale topology between lesion types.

Overall, these visualizations support the view that multipersistence encodes class-specific, multiscale topology that is coherent with known dermoscopic morphologies. While we do not claim these descriptors are directly diagnostic on their own, they offer an interpretable intermediate representation: the regions of the red–green grid where our Betti tensors are most active correspond to characteristic patterns of lesion fragmentation and hole formation that our TopoCon-MP model can exploit during training.

**t-SNE Visualization of Learned Embeddings**    We use t-distributed Stochastic Neighbor Embedding (t-SNE) to qualitatively examine the structure of learned feature representations on the DermaMNIST dataset. This visualization aims to provide intuition about class-wise clustering behavior in the embedding space.

We compare embeddings extracted from a frozen ImageNet-pretrained Swin Transformer (Tiny) with embeddings produced by a topology-augmented model that fuses Swin-T image features with multi-persistence (MP) topological descriptors.

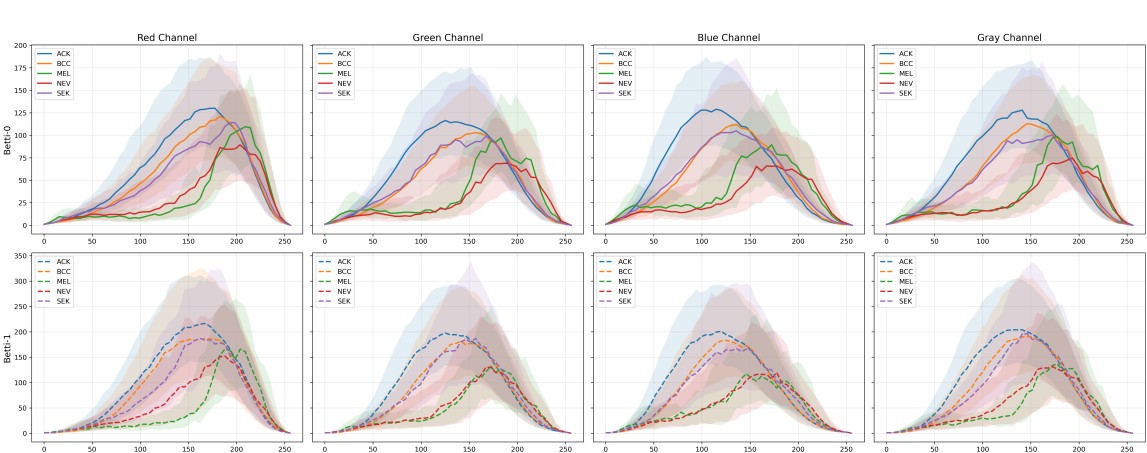

Figure 9: **Betti curves for PAD-UFES.** Mean Betti curves with 40% confidence bands for five MILK-10K lesion classes (BCC, NV, BKL, MEL, AKIEC) across color channels. Columns correspond to red, green, blue, and grayscale intensity filtrations. The top row shows $\beta_0$ (number of connected components) and the bottom row shows $\beta_1$ (number of holes) as functions of the threshold value. Solid (top) and dashed (bottom) lines indicate classwise means, and shaded regions mark the central 40% of subjects per class, revealing systematic differences in multiscale topology between lesion types.

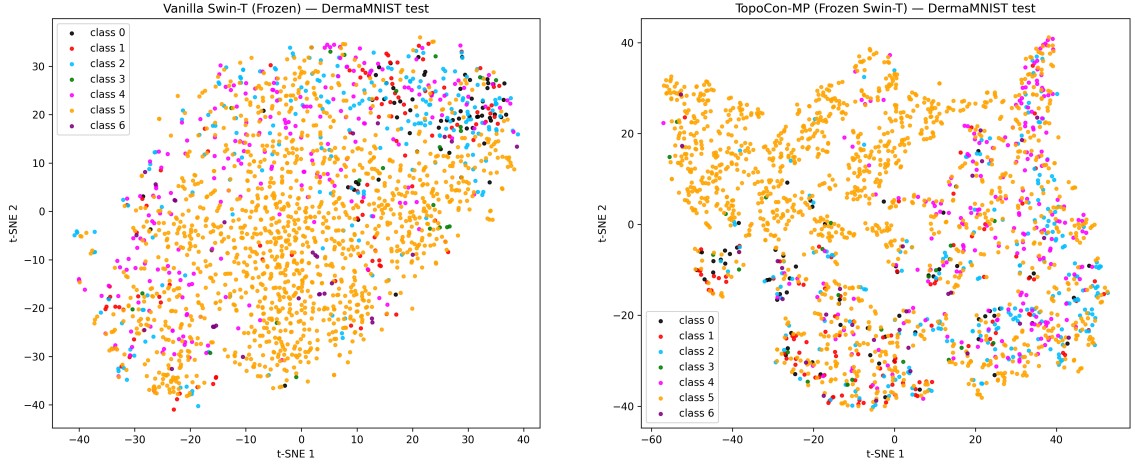

Figure 10: t-SNE visualization of DermaMNIST embeddings. **Left:** embeddings from a frozen Swin-T backbone. **Right:** topology-augmented embeddings obtained via TopoCon-MP. Colors indicate the seven diagnostic classes.

The topology-augmented embeddings exhibit more compact intra-class clusters and clearer separation between several diagnostic categories compared to the baseline Swin-

T representation. While t-SNE provides only a qualitative view of the embedding space, the observed clustering patterns suggest that incorporating multi-persistence topological information introduces complementary structural cues that enhance feature discrimination.

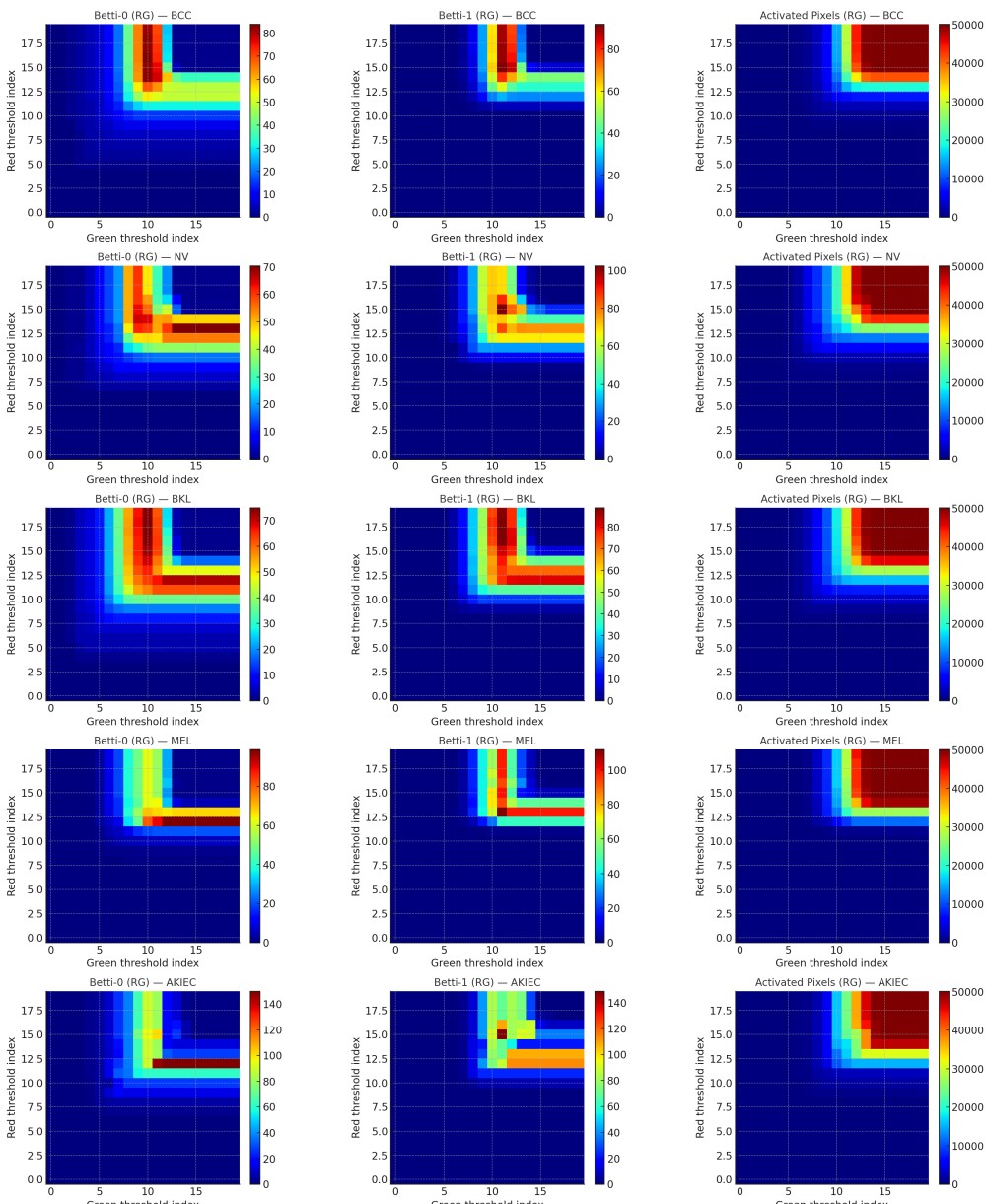

Figure 11: **Multipersistence heatmaps for MILK-10K.** Classwise median multiparameter descriptors computed on the red–green bifiltration grid ($20 \times 20$ thresholds). Rows correspond to lesion classes (BCC, NV, BKL, MEL, AKIEC) and columns show $\beta_0$, $\beta_1$, and activated-pixel counts, respectively. Color encodes the median value at each grid point, highlighting distinct patterns of connected components, holes, and overall lesion occupancy across classes in the multipersistence representation.

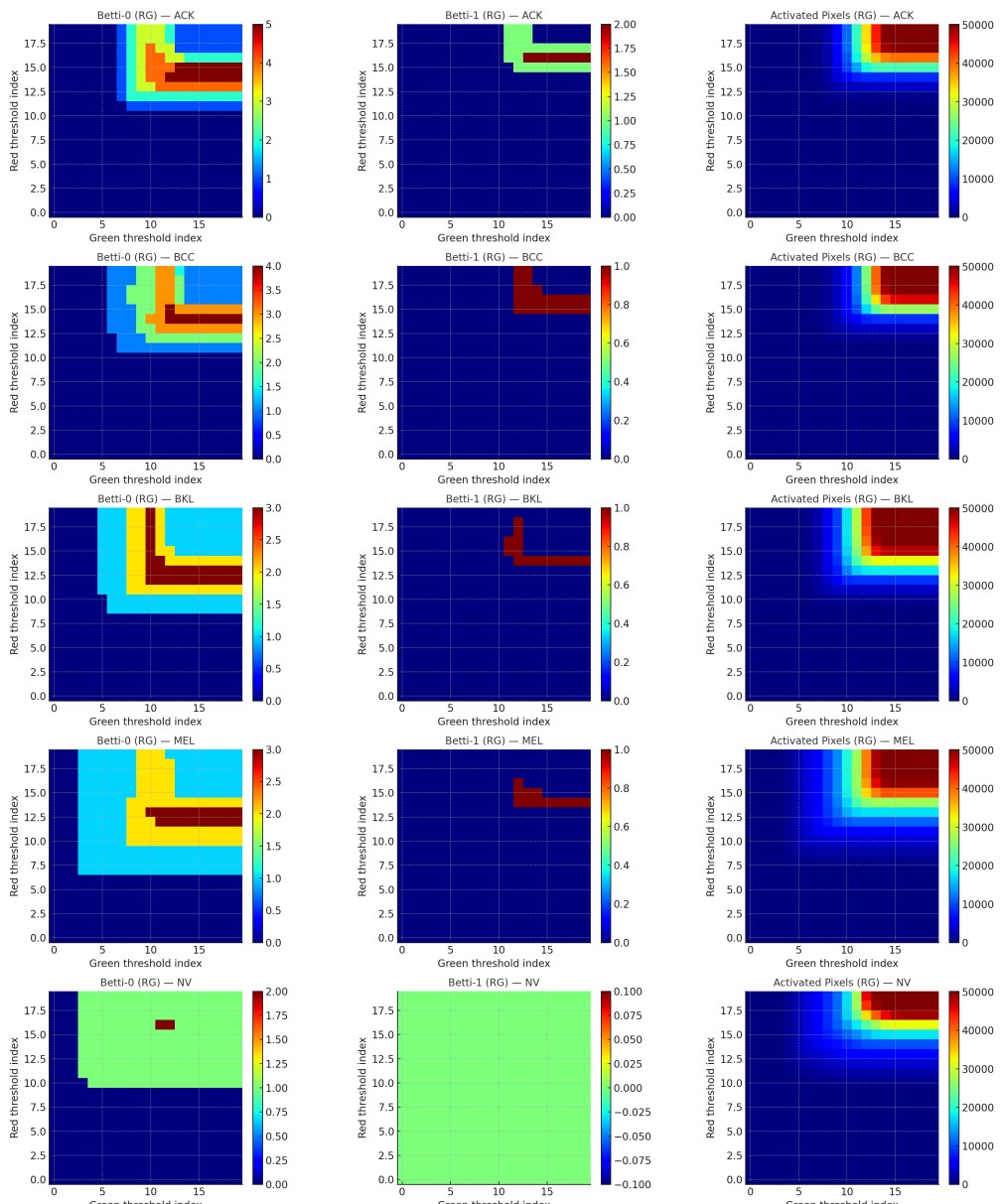

Figure 12: **Multipersistence heatmaps for DERMAMNIST.** Classwise median multiparameter descriptors computed on the red–green bifiltration grid ($20 \times 20$ thresholds). Rows correspond to lesion classes (BCC, NV, BKL, MEL, AKIEC) and columns show $\beta_0$, $\beta_1$, and activated-pixel counts, respectively. Color encodes the median value at each grid point, highlighting distinct patterns of connected components, holes, and overall lesion occupancy across classes in the multipersistence representation.

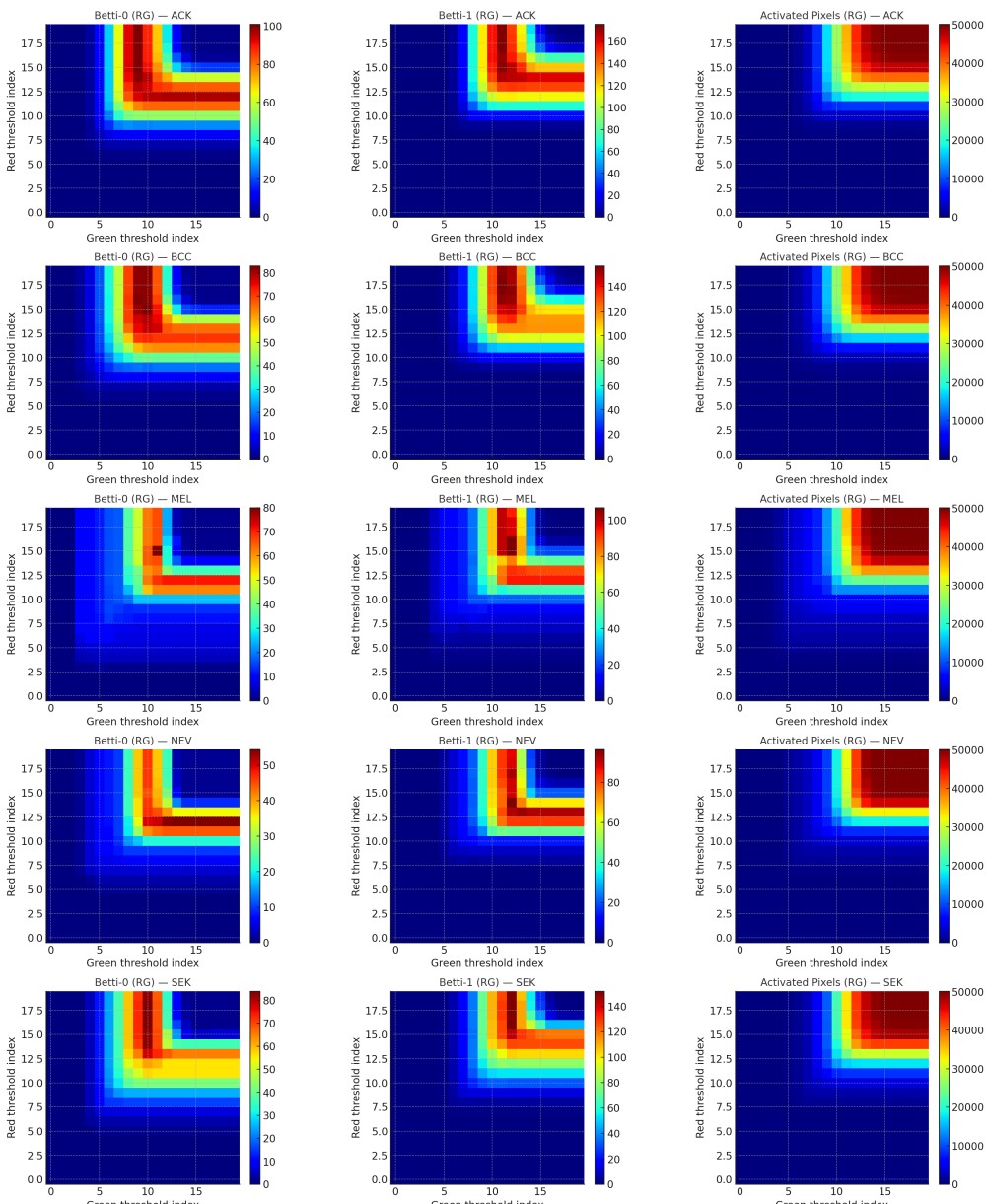

Figure 13: **Multipersistence heatmaps for PAD-UFES.** Classwise median multiparameter descriptors computed on the red–green bifiltration grid ($20 \times 20$ thresholds). Rows correspond to lesion classes (BCC, NV, BKL, MEL, AKIEC) and columns show $\beta_0$, $\beta_1$, and activated-pixel counts, respectively. Color encodes the median value at each grid point, highlighting distinct patterns of connected components, holes, and overall lesion occupancy across classes in the multipersistence representation.

