# OpenReview forum: "MultiPersistence Topological Fusion with Vision Transformers for Skin Cancer Detection"
_MIDL.io/2026/Conference — MIDL 2026 Poster_

### Official Review · Reviewer_AYiz · 2025-12-16

**Confidence:** 4
**Preliminary Rating:** 4
**Final Rating:** 4

**Summary:**

1. The paper proposes TopoCon-MP, a framework that integrates cubical multiparameter persistent homology (MP-PH) with a Vision Transformer (Swin-T) using a topology-aware supervised contrastive learning objective.
2. Dermoscopic images are transformed into 3×20×20 multipersistence maps (Betti-0, Betti-1, activated-pixels), derived from red-green bifiltrations, providing interpretable structural information.
3. These topological descriptors are encoded with an MLP and aligned with image embeddings using supervised contrastive learning, treating topology as a label-preserving secondary "view".
4. Ablations demonstrate that multipersistence and contrastive fusion both contribute, and visualizations reveal class-specific topology patterns reflecting clinically meaningful lesion morphology.

**Strengths:**

1. The paper is among the first to apply multiparameter persistent homology to dermoscopic image analysis, addressing known limitations of single-parameter TDA approaches. Prior work such as CuMPerLay (Korkmaz et al.) focuses on MP-PH representation learning but has not been applied to skin imaging.
2. The Betti tensor representation is computationally efficient compared to more complex MP-PH vectorizations (e.g. signed barcode measures (Loiseaux et al.)), and integrates seamlessly with transformer backbones.
3. The proposed topology-aware supervised contrastive learning idea instead of using heavy augmentations (unsafe in dermoscopy), the authors generate semantically consistent views via multipersistence.
4. The ablation studies are clearly structured and isolate (a) multipersistence vs. single-parameter summaries and (b) topology-only performance vs. fused contrastive learning.
5. The interpretability visualizations (Betti curves, multipersistence heatmaps) meaningfully support the claim that topology captures morphological lesion features.

**Weaknesses:**

1. The use of red-green bifiltration is empirically chosen; however, MP-PH is known to be sensitive to filtration selection. No sensitivity or alternative filtration experiments are included.
2. Betti tensors discard persistence lifetimes, limiting expressive power. More informative MP vectorizations (e.g., CuMPerLay, signed-barcode embeddings) are not compared against, making it difficult to assess whether Betti tensors are optimal.
3. No computational cost analysis is provided. MP-PH can be computationally expensive, and runtime or scaling information is essential for understanding applicability.
4. Performance gains over strong transformer baselines are sometimes modest (e.g., DermaMNIST). Without confidence intervals or statistical tests, it is unclear whether improvements are statistically significant.
5. The paper does not evaluate any standard contrastive learning approaches (SimCLR, MoCo, supervised contrastive learning with augmentations, or contrastive ViT pretraining). This is a significant gap because it prevents understanding whether improvements come from (a) multipersistence features, (b) supervised contrastive learning itself, or (c) the specific topology-image fusion strategy. No ablation is provided where the topology branch is replaced with a second augmented image, which is necessary to isolate the effect of topology.

**Detailed Comments:**

1. Please justify the filtration choice more clearly or provide sensitivity results for different filtration configurations (e.g., RGB bifiltration, intensity-erosion filtrations).
2. Include runtime and memory benchmarks for MP-PH computation. This is essential to evaluate the feasibility of multi-parameter TDA in practice.
3. Provide statistical significance testing (e.g., bootstrapped CI or paired tests) for AUC/F1 improvements.
4. Include variability across random seeds to assess stability.

**Justification Of Final Rating:**

After considering the authors' rebuttal, I lean toward a weak accept. The authors respond carefully and substantively to the main concerns, adding meaningful analyses that strengthen the paper. In particular, the sensitivity study on filtration parameters and channel selection demonstrates that TopoCon-MP is reasonably robust to practical design choices, addressing a key methodological concern. The added embedding visualizations help clarify the effect of topology, image alignment and provide useful qualitative support for the proposed objective. While some aspects, such as runtime scaling and statistical significance across seeds, remain somewhat limited or deferred to other sections, the authors are transparent about these limitations and position the work appropriately. Overall, the rebuttal increases confidence that the reported gains are not artifacts of specific parameter choices, and the method represents a technically sound and interesting contribution that merits acceptance despite remaining weaknesses.

**Justification Of The Preliminary Rating:**

1. The paper introduces integration of multiparameter persistent homology and transformer-based models, offering interpretable structural signal and improved performance.
2. The conceptual contribution, topology-aware supervised contrastive learning, is meaningful and appropriately motivated by the limitations of standard augmentations in dermoscopy.
3. Empirical results are strong overall, with consistent improvements across datasets and solid ablations demonstrating the necessity of both multipersistence and contrastive fusion.
4. The main weaknesses (lack of contrastive baselines, missing computational analysis, limited statistical evaluation) are important but addressable and do not undermine the core contribution.
5. The work is likely to stimulate further research in TDA-enhanced deep learning, making it worthy of acceptance, though not at the strongest level.

**Questions To Address In The Rebuttal:**

1. Why were no contrastive learning baselines (SimCLR, MoCo, supervised image-only SupCon) included, and can the authors provide at least an image-only SupCon baseline?
2. How sensitive is TopoCon-MP to the choice of filtration parameters and channel selection?
3. Can the authors report runtime or scaling characteristics for computing multipersistence features?
4. Are the performance improvements statistically significant across multiple seeds?
5. Does replacing the topology branch with a second augmented image reduce performance?
6. Would more expressive MP-PH vectorizations (e.g., CuMPerLay) potentially outperform Betti tensors?
7. Can the authors provide embedding visualizations illustrating topology-image alignment?

---

> ### Author Response · Authors · 2026-01-25
> **Responses to Reviewer Ayiz - part 1**
>
> *We thank the reviewer for the thoughtful and constructive feedback. Below we address each point in detail; a revised version of the manuscript incorporating these changes has been uploaded and is available as supplementary material in the rebuttal response above.*
>
> ### W1. MP Color choices
>  >The use of red-green bifiltration is empirically chosen; however, MP-PH is known to be sensitive to filtration selection. No sensitivity or alternative filtration experiments are included.
> > Please justify the filtration choice more clearly or provide sensitivity results for different filtration configurations (e.g., RGB bifiltration, intensity-erosion filtrations).
>
> Thank you for this important point. We agree that multiparameter PH can be sensitive to the filtration choice, and our red green bifiltration is an empirically motivated design decision. We chose a 2 parameter construction to keep MP PH computation feasible and stable in practice, since moving to three parameters or to many alternative filtrations quickly becomes substantially more expensive and is not yet supported by equally efficient pipelines. Within this 2 channel budget, we selected the pair that provided the most complementary signal on these dermoscopic datasets: red and green exhibited the lowest cross channel correlation and the clearest class separating behavior in the Betti curve trends, while blue tended to be more redundant. To make this rationale transparent, we expanded the Color multifiltrations section and added appendix visualizations that show the per channel Betti curve behavior across datasets. We also discuss filtration sensitivity as an explicit limitation and outline future work to evaluate modality specific filtrations such as intensity plus morphology based filtrations (e.g., erosion) and to design acquisition robust filtrations for broader deployment.
>
> ### W2.  Betti Tensors
>  > Betti tensors discard persistence lifetimes, limiting expressive power. More informative MP vectorizations (e.g., CuMPerLay, signed-barcode embeddings) are not compared against, making it difficult to assess whether Betti tensors are optimal.
> > Would more expressive MP-PH vectorizations (e.g., CuMPerLay) potentially outperform Betti tensors?
>
> Thank you for raising this. We agree that Betti tensors are a deliberate tradeoff between expressivity and practical learnability. In multiparameter persistence, stable and scalable vectorization is itself a major bottleneck, and more expressive approaches with learnable filtrations or complex MP embeddings typically require substantially more data and tuning to generalize well. Our goal here is to keep the MP representation simple and structured so that standard vision style learning can exploit it: the Betti tensor preserves the 2D bifiltration geometry as an “image like” object that integrates naturally with contrastive fusion, whereas many richer MP vectorizations discard this grid structure even if they retain lifetime information. Finally, our visual analysis (e.g., Fig. 7) suggests that discriminative signal is not dominated by a few large features; rather it is distributed across many small topological events (the overall curve density), which is well captured by Betti curve and tensor summaries. We have clarified this motivation in the revised manuscript (Appendix A) and note comparison to richer MP embeddings as future work.
>
> ### W3. Computational Time
> > No computational cost analysis is provided. MP-PH can be computationally expensive, and runtime or scaling information is essential for understanding applicability.
> > Include runtime and memory benchmarks for MP-PH computation. This is essential to evaluate the feasibility of multi-parameter TDA in practice.
>
>
> Thank you for raising this point. In the revised manuscript, we now report the computational cost of the multipersistence feature extraction, including both runtime and basic resource requirements. Multipersistence feature extraction on the DermaMNIST dataset was performed on an HPC system using single-core CPU execution. For 224×224 images, computing the 3×20×20 (Betti-0, Betti-1, activated pixels) tensor requires 0.229 s per image on average, corresponding to an overall preprocessing time of approximately 38 minutes for the full DermaMNIST dataset (10,015 images).Training TopoCon-MP with a frozen Swin-T backbone on the DermaMNIST dataset required approximately 55 minutes on an HPC system using a single NVIDIA A100 GPU. We added these details to the Implementation section and to the experimental setup so readers can assess the overhead and feasibility.

---

> > ### Author Response · Authors · 2026-01-25
> > **Responses to Reviewer Ayiz - part 2**
> >
> > ### W4. Statistical Significance
> > > Performance gains over strong transformer baselines are sometimes modest (e.g., DermaMNIST). Without confidence intervals or statistical tests, it is unclear whether improvements are statistically significant.
> > > Provide statistical significance testing (e.g., bootstrapped CI or paired tests) for AUC/F1 improvements.
> > Include variability across random seeds to assess stability.
> >
> > Thank you for this suggestion. In the original submission, we reported single runs due to the computational cost across multiple datasets and model variants, together with multipersistence (MP) feature extraction per split. Given the rebuttal page limits and runtime budget, we prioritized breadth of comparisons and ablations. We are currently rerunning all baselines and our model with 5 random seeds and will update Table 2 to report mean ± standard deviation. We will also report statistical significance across models using a paired test across seeds (including p-values) to confirm that the observed gains are consistent beyond run-to-run variation.
> >
> > ### W5. Other Contrastive Learning Approaches
> > > The paper does not evaluate any standard contrastive learning approaches (SimCLR, MoCo, supervised contrastive learning with augmentations, or contrastive ViT pretraining). This is a significant gap because it prevents understanding whether improvements come from (a) multipersistence features, (b) supervised contrastive learning itself, or (c) the specific topology-image fusion strategy. No ablation is provided where the topology branch is replaced with a second augmented image, which is necessary to isolate the effect of topology.
> >
> > Thank you for this important point. We agree that a standard contrastive baseline is needed to separate the effect of supervised contrastive learning from the effect of using topology as an additional view. Given the rebuttal time constraints, we implemented a representative image-only supervised contrastive learning baseline using standard augmentations (random cropping and horizontal flipping) and trained the same Swin-T backbone under the same training budget. We then compared it against the vanilla Swin-T baseline and our TopoCon-MP (topology–image SupCon) variant.
> >
> > On DermaMNIST, the image-only SupCon baseline does not improve over vanilla Swin-T, whereas TopoCon-MP yields the strongest overall performance, supporting that the gains are not simply due to using a supervised contrastive objective, but benefit from the topology view and topology–image alignment. We are also running other topology-image fusion strategies (late-fusion, concatenation), and will add them to the revision.
> >
> > | Model                       |   AUC |   Acc |    F1 |  Sens |  Spec |
> > | --------------------------- | ----: | ----: | ----: | ----: | ----: |
> > | Vanilla Swin-T              | 93.30 | 73.70 | 57.10 | 63.00 | 94.40 |
> > | SupCon Swin-T (crop + flip) | 91.13 | 68.13 | 48.02 | 58.76 | 94.04 |
> > | TopoCon-MP (Swin-T+MP)         | 94.90 | 79.30 | 62.20 | 59.70 | 94.30 |
> >
> > ### C1. Color choices
> > > Please justify the filtration choice more clearly or provide sensitivity results for different filtration configurations (e.g., RGB bifiltration, intensity-erosion filtrations).
> >
> > Please see our response to **Weakness 1.**
> >
> > ### C2. Runtime
> > > Include runtime and memory benchmarks for MP-PH computation. This is essential to evaluate the feasibility of multi-parameter TDA in practice.
> >
> > Please see our response to **Weakness 3.**
> >
> >
> > ### C3. Statistical Significance
> > >Provide statistical significance testing (e.g., bootstrapped CI or paired tests) for AUC/F1 improvements.
> > Include variability across random seeds to assess stability.
> >
> > Please see our response to **Weakness 4.**
> >
> > ### Q1. No contrastive learning.
> > > Why were no contrastive learning baselines (SimCLR, MoCo, supervised image-only SupCon) included, and can the authors provide at least an image-only SupCon baseline?
> >
> > Please see our response to **Weakness 5.**

---

> > > ### Author Response · Authors · 2026-01-25
> > > **Responses to Reviewer Ayiz - part 3**
> > >
> > > ### Q2. the choice of filtration parameters and channel
> > > > How sensitive is TopoCon-MP to the choice of filtration parameters and channel selection?
> > >
> > > Thank you for this question. Filtration design can affect MP PH features, so we added an explicit sensitivity analysis in Appendix B. First, we vary the filtration discretization by changing the number of intensity thresholds in single parameter persistence on MILK 10K from 50 to 100 to 250 (feature dimension 600 to 1200 to 3000) while keeping the downstream classifier fixed.
> > >
> > > As shown in Appendix B, performance is stable across this range with only modest variation in AUC, accuracy, F1, sensitivity, and specificity, indicating limited sensitivity to threshold resolution once a reasonable discretization is used. Second, we study channel selection by computing single channel persistence features from the red, green, and blue channels (same feature budget) on DermaMNIST, MILK 10K, and PAD. Appendix B shows that channel rankings can vary by dataset but overall performance is comparable, suggesting that no single channel is universally dominant. These results motivate our use of complementary channels in the multipersistence construction and support that the topological descriptors are reasonably robust to practical choices of discretization and channel selection.
> > >
> > >
> > > | Dataset  | # thresholds | # features (all channels) |  AUC |  Acc |   F1 | Sens | Spec |
> > > | -------- | -----------: | ------------------------: | ---: | ---: | ---: | ---: | ---: |
> > > | MILK-10K |           50 |              50×3×4 = 600 | 74.9 | 59.5 | 19.6 | 19.8 | 93.8 |
> > > | MILK-10K |          100 |            100×3×4 = 1200 | 74.8 | 60.1 | 20.4 | 20.4 | 94.0 |
> > > | MILK-10K |          250 |            250×3×4 = 3000 | 74.5 | 60.4 | 21.5 | 21.2 | 94.0 |
> > >
> > > ### DermaMNIST
> > >
> > > | TDA model | # Features |  AUC |  Acc |   F1 | Sens | Spec |
> > > | --------- | ---------: | ---: | ---: | ---: | ---: | ---: |
> > > | SP_Red    |        150 | 83.4 | 70.0 | 28.2 | 25.6 | 88.9 |
> > > | SP_Green  |        150 | 85.4 | 71.2 | 31.6 | 28.5 | 89.9 |
> > > | SP_Blue   |        150 | 87.7 | 71.0 | 35.6 | 31.6 | 90.5 |
> > >
> > > ### MILK-10K
> > >
> > > | TDA model | # Features |  AUC |  Acc |   F1 | Sens | Spec |
> > > | --------- | ---------: | ---: | ---: | ---: | ---: | ---: |
> > > | SP_Red    |        150 | 71.5 | 57.4 | 17.3 | 18.0 | 93.4 |
> > > | SP_Green  |        150 | 75.6 | 57.6 | 17.9 | 18.3 | 93.4 |
> > > | SP_Blue   |        150 | 72.3 | 57.6 | 17.6 | 18.3 | 93.6 |
> > >
> > > ### PAD
> > >
> > > | TDA model | # Features |  AUC |  Acc |   F1 | Sens | Spec |
> > > | --------- | ---------: | ---: | ---: | ---: | ---: | ---: |
> > > | SP_Red    |        150 | 71.1 | 44.7 | 23.5 | 25.0 | 86.2 |
> > > | SP_Green  |        150 | 72.3 | 47.8 | 27.7 | 29.0 | 87.0 |
> > > | SP_Blue   |        150 | 73.7 | 51.3 | 31.4 | 32.2 | 87.8 |
> > >
> > > ### Q3. Runtime
> > > Can the authors report runtime or scaling characteristics for computing multipersistence features?
> > >
> > > Please see our response to **Weakness 3.**
> > >
> > >
> > > ### Q4. Statistical significance
> > > Are the performance improvements statistically significant across multiple seeds?
> > >
> > > Please see our response to **Weakness 4.**
> > >
> > >
> > > ### Q5. Using another augmented image
> > > > Does replacing the topology branch with a second augmented image reduce performance?
> > >
> > > Please see our response to **Weakness 4.**
> > >
> > > ### Q6. Other MP vectorizations.
> > > > Would more expressive MP-PH vectorizations (e.g., CuMPerLay) potentially outperform Betti tensors?
> > >
> > > Please see our response to **Weakness 2.**
> > >
> > > ### Q7. Visualizations for Topology-Image alignment
> > > > Can the authors provide embedding visualizations illustrating topology-image alignment?
> > >
> > >
> > > Thank you for the suggestion. In the revised manuscript, we added embedding visualizations to illustrate the effect of topology image alignment: we report t SNE plots of the learned image embeddings for a vanilla Swin T baseline and for TopoCon MP on DermaMNIST. The TopoCon MP embeddings show clearer class-wise clustering and separation compared to the vanilla model, which is consistent with the alignment objective encouraging image representations to better reflect the complementary structural cues captured by the multipersistence branch. These visualizations are included in the appendix.

---

### Official Review · Reviewer_42W2 · 2026-01-02

**Confidence:** 5
**Preliminary Rating:** 3
**Final Rating:** 4

**Summary:**

The authors propose a hybrid medical‑AI pipeline for skin‑lesion classification that fuses topological data analysis (TDA) with a lightweight vision‑transformer backbone.

Their main hypothesis is that the integration of TDA provides interpretable shape‑based cues, while MobileViT offers a mobile‑ready solution suitable for point‑of‑care diagnostics. The results reportedly outperform baseline CNNs on the chosen datasets, with statistically significant gains in AUC.

**Strengths:**

The work tackles a relevant problem (skin‑cancer screening) and employs a rigorous methodological stack: TDA, transformer architecture, and knowledge distillation.

Utilizing TDA information to enhance the AI model's diagnostic decision-making mimics clinicians' analysis.

The experimental design appears thorough (cross‑validation, ablation studies).

The presentation is generally clear. Figures summarizing barcodes and model architecture are helpful

Combining TDA with a mobile‑friendly transformer is novel and could bridge the gap between research prototypes and deployable clinical tools.

The use of publicly available datasets enhances reproducibility.

**Weaknesses:**

Robustness of the proposed method against artifacts in the images

Comparing against the models (baseline CNN and ViT) does not seem fair.

No comparison with the transformer-only model trained using MP-SupCon alignment is provided.

Some sub‑figures lack captions or axis labels. Figures 3 & 4 are not refered in the text.

Similarly, the takeway message from Figure 5 is not clear.

Some conclusions are not supported by experimental data

**Detailed Comments:**

As it is presented in the paper, the persistence homology model utilizes a threshold on pixel intensity values to extract topology information. Color calibration and brightness variations are a very well-known issue in dermatology images. In this respect, it is not clear how robust the PH method is under varying lighting conditions.

**Justification Of Final Rating:**

Thanks to the authors for their comprehensive answers to the points raised in the review.
They updated the article based on the review comments and also committed to making a few more changes to the camera-ready paper to follow up on the rebuttal discussions.
I am raising my score to weak accept.

**Justification Of The Preliminary Rating:**

The idea of utilizing topological information in decision making process is good.
Topology information extraction method is well presented

The presentation lacks an analysis on the robustness of the methods against imaging artifacts and variation of lighting conditions.
Some of the figures are not clear.
Missing comparisons.

**Questions To Address In The Rebuttal:**

Did the authors use any image augmentation methods to increase the robustness of the method against intensity variations, making it shift invariant, etc...

It is not clear what the authors mean by "augmentation induced bias" on page 6, end of the first paragraph.

In Figure 3, doesn't the TopoCon-MP module take both topology and image representations as input?

The baseline model backbones used for comparisons are frozen, whereas the Swin Transformer backbone in the proposed model is not. Do you think this leads to a fair comparison?

The proposed TopoCon-MP loss aims to align the Swin Transformer embeddings with the Topology portion of the model. Can the authors present a comparison between the overall model and a Swin-Transformer-only model after the alignment?

In the conclusions section, the authors state that the topological analysis presented in the paper supports transparent decision-making. However, no data is presented in this respect. Yes, the reviewer agrees that the presented topology-based information extraction methods are interpretable; however, it is unclear how this leads to more transparent decision-making. Can the authors present concrete examples?

---

> ### Author Response · Authors · 2026-01-25
> **Response to Reviewer 42W2 - part 1**
>
> We thank the reviewer for the thoughtful and constructive feedback. Below we address each point in detail; a revised version of the manuscript incorporating these changes has been uploaded and is available as supplementary material in the rebuttal response above.
>
> ### W1. Robustness
> > Robustness of the proposed method against artifacts in the images.
> > As it is presented in the paper, the persistent homology model utilizes a threshold on pixel intensity values to extract topology information. Color calibration and brightness variations are a very well-known issue in dermatology images. In this respect, it is not clear how robust the PH method is under varying lighting conditions.*
>
> Thank you for raising this robustness concern. We agree that absolute intensity thresholds can be sensitive to lighting and color calibration shifts, particularly in cross-domain transfer settings. In standard within-dataset evaluations, preprocessing and normalization already mitigate some of this sensitivity; however, domain generalization remains challenging for these variations.
>
> To directly assess robustness to brightness variations, we replaced the fixed intensity threshold grid used for multipersistence persistent homology (MP-PH) with a dataset-specific quantile-based grid, using 50 quantiles instead of 50 fixed intensity thresholds. This modification makes the filtration invariant to monotone intensity rescaling and is therefore more robust under global brightness shifts.
>
> Using the same cross-dataset transfer setting as in Table 3 and the same XGBoost classifier on the 600-dimensional TDA features, we obtain the following results:
>
> | Thresholding                      | AUC (macro) |   Acc | F1 (macro) |  Sens |  Spec |
> | --------------------------------- | ----------: | ----: | ---------: | ----: | ----: |
> | Vanilla 50 (fixed thresholds)     |       63.43 | 36.08 |      26.65 | 27.27 | 82.40 |
> | Adjusted 50 (quantile thresholds) |       60.61 | 30.38 |      24.53 | 28.17 | 82.11 |
>
> In this transfer experiment, quantile-based thresholding yields similar sensitivity and specificity and comparable overall behavior, although it does not improve AUC or F1. We will include this analysis in the revised manuscript as a robustness control and discuss it accordingly. We also outline future work on acquisition-robust filtrations and optional color calibration normalization to further improve robustness under varying imaging conditions.
>
> ### W2. SOTA Comparisons
> > Comparing against the models (baseline CNN and ViT) does not seem fair. The baseline model backbones used for comparisons are frozen, whereas the Swin Transformer backbone in the proposed model is not. Do you think this leads to a fair comparison?
>
> Thank you for raising this fairness concern. In the original submission, all methods, including TopoCon-MP, were evaluated under the same compute-constrained protocol in which ImageNet-pretrained backbones were kept frozen and only the task-specific heads and lightweight fusion layers were trained. We adopted this linear-probe-style setting to control model capacity, reduce overfitting on small medical datasets, and isolate the contribution of the multipersistence branch and alignment objective. Under this protocol, the comparison is fair, as the backbone training status was identical across all baselines and our method.
>
> For the rebuttal revision, we are now rerunning all methods with end-to-end trainable backbones under a matched training budget and repeating each experiment over five random seeds.
>
> ### W3. No Transformer-Only Model
> > No comparison with the transformer-only model trained using MP-SupCon alignment is provided. The proposed TopoCon-MP loss aims to align the Swin Transformer embeddings with the topology portion of the model. Can the authors present a comparison between the overall model and a Swin-Transformer-only model after the alignment?
>
> Thank you for this comment. In Table 2, Swin-T already serves as the transformer-only baseline, using the same Swin-T backbone and training protocol as TopoCon-MP but without any topological branch (i.e., image-only cross-entropy training). We agree, however, that this baseline does not isolate the effect of the MP-SupCon alignment objective itself. In the revised version, we therefore add an additional image-only SupCon control, where Swin-T is trained using the same supervised contrastive objective (with two augmented image views) and subsequently fine-tuned for classification. This enables a direct comparison between TopoCon-MP and a Swin-T model “after alignment” but without topology, thereby isolating the contribution of the multipersistence alignment mechanism.
>
> ### W4. Subfigure Captions
> > Some sub-figures lack captions or axis labels. Figures 3 and 4 are not referred to in the text.*
>
> Thank you for catching this. In the revised manuscript, we have added the missing subfigure captions, and we now explicitly reference Figures 3 and 4 in the main text.

---

> > ### Author Response · Authors · 2026-01-25
> > **Response to Reviewer 42W2 - part 2**
> >
> > ### W5. Figure 5
> > > Similarly, the takeaway message from Figure 5 is not clear.
> >
> > Thank you for pointing this out. The intent of Figure 5 is to provide an interpretable, class-level view of how lesion topology evolves under the red-channel sublevel filtration. Different classes exhibit distinct Betti-0 trajectories and peak locations, reflecting differences in the number and scale of connected components across intensity thresholds. From a machine learning perspective, each curve is discretized into a fixed 50-dimensional embedding per image; thus, even modest but consistent differences across thresholds accumulate into separations in $\mathbb{R}^{50}$ that help distinguish classes. We have revised the surrounding text to make this takeaway explicit and to clarify how these signatures motivate our multipersistence-based fusion.
> >
> > ### W6. Conclusions
> > > Some conclusions are not supported by experimental data. In the conclusions section, the authors state that the topological analysis presented in the paper supports transparent decision-making. However, no data is presented in this respect. While the topology-based information extraction methods are interpretable, it is unclear how this leads to more transparent decision-making. Can the authors present concrete examples?
> >
> > Thank you for this important point. We agree that the original wording was too strong. While the proposed topological descriptors are intrinsically interpretable, the original submission did not provide concrete evidence directly linking them to more transparent decision-making. In the revised manuscript, we add qualitative case studies that present the input image, the predicted class, and the corresponding multipersistence maps and Betti curves. These examples illustrate how class-specific topological patterns align with clinically meaningful structures and can be inspected alongside the model output as an auxiliary form of explanation. We also revise the Conclusions section to state this more precisely, framing topology as an interpretable complementary view that supports model auditing rather than a standalone guarantee of transparency.
> >
> > ### C1. PH Usefulness
> > > As it is presented in the paper, the persistent homology model utilizes a threshold on pixel intensity values to extract topology information. Color calibration and brightness variations are a very well-known issue in dermatology images. In this respect, it is not clear how robust the PH method is under varying lighting conditions.
> >
> > Please see our response to **W1 (Robustness)**.
> >
> > ### Q1. Augmentation
> > > Did the authors use any image augmentation methods to increase the robustness of the method against intensity variations, making it shift invariant, etc.?
> >
> > No explicit image augmentation was applied in our experiments. Images were resized and normalized using standard preprocessing procedures.
> >
> > ### Q2. Augmentation-Induced Bias
> > > It is not clear what the authors mean by “augmentation induced bias” on page 6, end of the first paragraph.
> >
> > Thank you for pointing this out. By “augmentation-induced bias,” particularly in medical imaging, we refer to the effect that strong contrastive augmentations (such as heavy color jitter, aggressive cropping, or blur) can have in distorting clinically meaningful cues and producing label-inconsistent views. This can encourage invariances that are undesirable for diagnostic tasks. We have clarified this wording in the revised manuscript and explicitly stated which augmentations we avoid and the rationale for doing so.
> >
> > ### Q3. TopoCon-MP
> > > In Figure 3, doesn't the TopoCon-MP module take both topology and image representations as input?
> >
> > Thank you for pointing this out. Yes, TopoCon-MP uses both the image embedding and the topology embedding as inputs. The topology branch encodes the multipersistence maps, and the alignment loss is computed between the two representations. We have updated Figure 3 to make these inputs and connections explicit.
> >
> > ### Q4. Frozen Backbones and Fairness
> > > The baseline model backbones used for comparisons are frozen, whereas the Swin Transformer backbone in the proposed model is not. Do you think this leads to a fair comparison?
> >
> > Please see our response to **W2 (SOTA Comparisons)**.
> >
> > ### Q5. Swin-Only Comparison
> > >The proposed TopoCon-MP loss aims to align the Swin Transformer embeddings with the topology portion of the model. Can the authors present a comparison between the overall model and a Swin-Transformer-only model after the alignment?
> >
> > Please see our response to **W3 (No Transformer-Only Model)**.
> >
> > ### Q6. Conclusions
> >
> > Please see our response to **W6 (Conclusions)**.

---

> > > ### Comment · Reviewer_42W2 · 2026-01-31
> > >
> > > I would like to thank the reviewers for their rebuttal responses. The addition of the new information and the changes made in the article, improved the presentation and the clarity.
> > >
> > > One issue still not clear on my end is W3 and how the model presented in Figure 4 come into play
> > >
> > > **W3. No Transformer-Only Model:**
> > > _No comparison with the transformer-only model trained using MP-SupCon alignment is provided. The proposed TopoCon-MP loss aims to align the Swin Transformer embeddings with the topology portion of the model. Can the authors present a comparison between the overall model and a Swin-Transformer-only model after the alignment?_
> > >
> > > As it is presented in the current version, the Topology branch (lower branch in Figure 3) is not trainable (reads like the number of levels is predetermined) during the Topocon-MP training. The calculation of Betti Tensors (Multiparameter Persistent Topological Features) is deterministic, so no backpropagation takes place over this branch.  During TopoMP-CON training, the model aligns the Swin-T model with Topological features (the self-supervised branch's task) while maximizing classification performance (the classification branch). In a sense, the self-supervised branch limits the model's ability to learn additional information beyond topological features unless there is some weighting in the loss, as presented in Figure 4. But the figure creates confusion, as it is not clear whether the "Outputs" here are projections used to align the 2 branches or are used to calculate the loss shown in the same figure.
> > > Also in the appendix, the authors gave a T-SNE plot of the Swin-T embeddings after TopoMP-Con training and show that the classes are more separated after the training. But this is kind of expected as the model is trained using class labels. It would be better to see the classification performance with these embeddings and compare it against the TOPO+Swin-T concatenated embeddings (TOPOMP-Con model in the main tables)

---

> > > > ### Author Response · Authors · 2026-01-31
> > > >
> > > > Thank you for the follow-up and for pinpointing the ambiguity around W3 and Figure 4. We agree our previous explanation was not sufficiently clear, and we clarify the training signals and provide a quantitative transformer-only control below.
> > > >
> > > > ### What is deterministic and what is trainable
> > > >
> > > > You are correct that the multipersistence computation itself is **deterministic**. The Betti tensors are produced by a fixed pipeline (red green bifiltration with a predetermined discretization), so **no backpropagation** happens into the multipersistence computation.
> > > >
> > > > However, the topology branch is not purely fixed. After computing the **3×20×20** multipersistence maps, we pass them through a **trainable topology encoder** to obtain a topology embedding. This topology encoder is trained jointly with the image side through:
> > > >
> > > > * the topology image supervised contrastive **alignment** loss
> > > > * the downstream **classification** objective (via fusion)
> > > >
> > > > So, while MP feature extraction is fixed, the mapping from MP maps to a representation used for alignment and fusion is **learned**.
> > > >
> > > >
> > > > ### What Figure 4 “Outputs” mean
> > > >
> > > > Figure 4 is meant to illustrate only the training time alignment mechanism. The “Outputs” are projected embeddings produced by small projection heads and used only to compute the supervised contrastive alignment loss between the image and topology views. They are not classification logits. Classification uses the fused representation shown in Figure 3. We will revise Figure 4 to rename these as “projected embeddings for SupCon” and explicitly annotate that they are used only for alignment and discarded at inference.
> > > >
> > > > ### Quantitative transformer only control beyond t-SNE
> > > >
> > > > We agree that the SNE separation alone is not sufficient, since class separation is expected in supervised settings. To isolate whether gains come simply from supervised contrastive learning, we ran an image-only SupCon baseline (two augmented image views: crop and flip) and report classification metrics below.
> > > >
> > > > | Model                                  |   AUC |   Acc |    F1 |  Sens |  Spec |
> > > > | -------------------------------------- | ----: | ----: | ----: | ----: | ----: |
> > > > | Vanilla Swin T                         | 93.30 | 73.70 | 57.10 | 63.00 | 94.40 |
> > > > | Image only SupCon Swin T (crop + flip) | 91.13 | 68.13 | 48.02 | 58.76 | 94.04 |
> > > > | TopoCon MP (Swin T plus MP)            | 94.90 | 79.30 | 62.20 | 59.70 | 94.30 |
> > > >
> > > > This shows that standard image only SupCon does not improve over vanilla Swin T in this dermoscopy setting, while TopoCon MP achieves the strongest overall AUC, accuracy, and F1, supporting that the gains are not due to the contrastive objective alone but to using topology as a complementary view.
> > > >
> > > > ### The transformer only after the topology alignment comparison you requested
> > > >
> > > > To address W3 directly, we will also report Swin T only performance after TopoCon MP training by discarding the topology branch and fusion at inference and evaluating the learned Swin T embeddings with a classifier head. We will present this aligned Swin T result alongside the three rows above to quantify how much improvement comes from topology-guided alignment shaping the image representation versus keeping topology at inference.
> > > >
> > > > Thank you very much again for your insightful comments and valuable feedback. We will update the figure and captions to make the role of the projection outputs and the alignment loss unambiguous and include the aligned Swin T classification control to resolve W3 quantitatively.

---

> ### Author Response · Authors · 2026-02-02
> **Thank you**
>
> Thank you for the constructive feedback and follow-up. We greatly appreciate the updated score and will incorporate the remaining changes in the camera-ready version as discussed.

---

### Official Review · Reviewer_e48T · 2026-01-08

**Confidence:** 2
**Preliminary Rating:** 4

**Summary:**

The authors present a method for extracting and combining topological features with vision transformers to detect deeper, multi-scale features for skin cancer detection. The authors perform a comprehensive analysis, using three publicly available datasets, compare against several state-of-the-art methods, and perform various ablation studies. This manuscript is well written in general. Minor improvements could be made by performing a statistical analysis and expanding on limitations and future work.

**Strengths:**

1. The authors provide a clear list of their contributions.
2. The authors provide comprehensive background information, including a discussion of other state-of-the-art ML algorithms for skin cancer detection and topological ML for medical image analysis.
3. The authors utilized three publicly available datasets for their analysis.
4. The authors compared their method against several other state-of-the-art algorithms.
5. The authors perform two ablation studies, one for topological feature ablation and another comparing different ML models.
6. The reviewers appreciate that the authors have made their code publicly available.

**Weaknesses:**

1. One limitation the authors provide is the “computational overhead of multipersistence”. It would be useful to provide some insight into the computation time required.
2. The limitations and future work in general could be expanded.
3. It would be useful to include some statistical analysis.
4. It would be nice to provide some standard deviation values in Tables 3 and 4.

**Detailed Comments:**

Extended comments:
1. Please provide any computational time and requirements for the multipersistence computation.
2. It would be nice to expand the limitations and future work. For instance, what issues do you foresee if using a similar approach on images other than dermascope images?
3. It would be useful to include some statistical analysis. For instance, are there statistically significant differences among the three models/features in Tables 3 and 4?
4. Standard deviation values should be included in Tables 3 and 4.

Minor comments:
1. Please comment on why, in the section “Color multifiltrations for dermoscopic images”, only the red and green channels were used and not blue?
2. The references to images in the appendix in the main paper should be removed. (Figures 7 and 8).
3. The sensitivity values for the DermaMNIST and MILK-10K in Table 2 are fairly low (59.7 and 50.4). Please comment on this.

**Justification Of The Preliminary Rating:**

The authors have performed an extensive analysis using multiple publicly available datasets, comparison against state-of-the-art methods, and also an inclusion of ablation studies. The manuscript is written well overall. Several improvements could be made, including statistical analysis and expansion on limitations and future work.

**Questions To Address In The Rebuttal:**

1. Could the authors please comment on statistical significance across models, and across features for Tables 3 and 4?
2. Could the authors please expand on the limitations and future work?

---

> ### Author Response · Authors · 2026-01-23
> **Response to Reviewer e48T - part 1**
>
> We thank the reviewer for the thoughtful and constructive feedback. Below we address each point in detail; a revised version of the manuscript incorporating these changes has been uploaded and is available as supplementary material in the rebuttal response above.
>
> ### W1. Computational Time
> >The limitation the authors provide is the “computational overhead of multipersistence”. It would be useful to provide some insight into the computation time required. Please provide any computational time and requirements for the multipersistence computation.
>
> Thank you for raising this point. In the revised manuscript, we now report the computational cost of the multipersistence feature extraction, including both runtime and basic resource requirements. Multipersistence feature extraction on the DermaMNIST dataset was performed on an HPC system using single-core CPU execution. For 224×224 images, computing the 3×20×20 (Betti-0, Betti-1, activated pixels) tensor requires 0.229 s per image on average, corresponding to an overall preprocessing time of approximately 38 minutes for the full DermaMNIST dataset (10,015 images).Training TopoCon-MP with a frozen Swin-T backbone on the DermaMNIST dataset required approximately 55 minutes on an HPC system using a single NVIDIA A100 GPU. We added these in the revision (Implementation Details).
>
> ### W2. Limitations
> >The limitations and future work in general could be expanded. It would be nice to expand the limitations and future work. For instance, what issues do you foresee if using a similar approach on images other than dermascope images?
>
> Thank you for the suggestion. In the revised manuscript, we have expanded the **Limitations and Future Work** paragraph (before the Conclusion) to note that extending TopoCon-MP beyond dermoscopy images may require modality-specific filtrations and preprocessing steps to ensure stability and meaningful topological representations. We also outline future directions focused on analyzing filtration sensitivity and developing acquisition-robust designs to support broader applicability across different medical imaging modalities.
>
> ### W3. Statistical Analysis
> > It would be useful to include some statistical analysis. For instance, are there statistically significant differences among the three models/features in Tables 3 and 4?
>
> Thank you for this suggestion. In the original submission, we reported single runs due to the computational cost across multiple datasets and model variants, together with multipersistence (MP) feature extraction per split. Given the rebuttal page limits and runtime budget, we prioritized breadth of comparisons and ablations. We are currently rerunning all baselines and our model with 5 random seeds and will update Table 2 to report mean ± standard deviation. We will also report statistical significance across models  using a paired test across seeds (including p-values) to confirm that the observed gains are consistent beyond run-to-run variation.
>
> ### W4. Standard Deviations
> > It would be nice to provide some standard deviation values in Tables 3 and 4. Standard deviation values should be included in Tables 3 and 4.
>
> Thank you for the suggestion. As mentioned in W3, in the original submission, we reported single runs due to the computational budget, but we are now rerunning all models across all datasets with 5 random seeds and will update Tables to report mean ± standard deviation for all reported metrics, providing a clearer view of performance variability and robustness.

---

> > ### Author Response · Authors · 2026-01-25
> > **Response to Reviewer e48T - part 2**
> >
> > ### C1. Why Red–Green Multipersistence
> > > Please comment on why, in the section “Color multifiltrations for dermoscopic images”, only the red and green channels were used and not blue.
> >
> > We use a two-parameter multipersistence construction to keep multipersistence persistent homology (MP-PH) computation practical at scale, as extending to three parameters is substantially more expensive and is not yet supported by equally efficient and stable computational pipelines. Within this two-channel budget, we selected the channel pair that provided the most complementary signal empirically. Across datasets, the red and green channels exhibited the lowest correlation and showed the clearest class-separating behavior in the resulting Betti curves, whereas the blue channel was more redundant and contributed less additional topological variation. To make this choice transparent, we have added Betti curve visualizations for all datasets in the Appendix and clarified both the computational motivation and the empirical channel selection rationale in the “Color multifiltrations for dermoscopic images” section.
> >
> > ### C2. References to Appendix
> > > The references to images in the appendix in the main paper should be removed (Figures 7 and 8).
> >
> > Thank you for pointing this out. In the revised manuscript, we have removed the direct references in the main text to Appendix Figures 7 and 8 and instead refer readers more generally to the supplementary material for additional qualitative examples.
> >
> > ### C3. Low Sensitivity
> > >The sensitivity values for the DermaMNIST and MILK-10K in Table 2 are fairly low (59.7 and 50.4). Please comment on this.
> >
> > Thank you for noting this. Sensitivity is the most challenging metric on DermaMNIST and MILK-10K due to class imbalance and the presence of clinically ambiguous cases evaluated at a fixed decision threshold. As a result, methods that improve overall discrimination (e.g., AUC and F1) can still exhibit modest recall. Importantly, TopoCon-MP achieves the highest sensitivity among all compared models on both datasets (59.7 on DermaMNIST and 50.4 on MILK-10K), while also improving AUC and F1, indicating that the observed gains are not driven by a specificity-only tradeoff. We have added a brief discussion of this point in the revised Results section.
> >
> > ### Q1*. Statistical Significance
> > >Could the authors please comment on statistical significance across models, and across features for Tables 3 and 4?
> >
> > Please see our response to **W3 (Statistical Analysis)**.
> >
> > ### Q2*. Limitations
> > >Could the authors please expand on the limitations and future work?
> >
> > Please see our response to **W2 (Limitations)**.

---

### Author Rebuttal · Authors · 2026-01-25

**Rebuttal:**

We thank the reviewers for their insightful and constructive feedback. We have uploaded a revised version of the paper as supporting material; all changes are highlighted in red. Below, we address each reviewer’s comments and questions point by point. We greatly appreciate your valuable time and feedback.

**Supporting Material:**

/attachment/c36a26ddb11a15bd82f2cefaad3ac7e7dbf472de.pdf

---

### Meta-Review · Area_Chair_GCXn · 2026-02-13

**Recommendation:** Accept (Poster)
**Confidence:** 5

**Metareview:**

clear acceptance

---

### Decision · Program_Chairs · 2026-02-13

Accept (Poster)